# Improving Receptor-Mediated Intracellular Access and Accumulation of Antibody Therapeutics—The Tale of HER2

**DOI:** 10.3390/antib9030032

**Published:** 2020-07-13

**Authors:** Jeffrey V. Leyton

**Affiliations:** Department of Nuclear Medicine and Radiobiology, Faculty of Medicine and Health Sciences, Centre Hospitalier Universitaire de Sherbrooke (CHUS), Université de Sherbrooke, Sherbrooke, QC J1H5N4, Canada; jeffrey.leyton@usherbrooke.ca; Tel.: +1-819-346-1110

**Keywords:** HER2, trastuzumab, trastuzumab-emtansine, antibodies, antibody–drug conjugates, internalization, accumulation, biopharmaceuticals, cost-effective, receptor crosslinking

## Abstract

Therapeutic anti-HER2 antibodies and antibody–drug conjugates (ADCs) have undoubtedly benefitted patients. Nonetheless, patients ultimately relapse—some sooner than others. Currently approved anti-HER2 drugs are expensive and their cost-effectiveness is debated. There is increased awareness that internalization and lysosomal processing including subsequent payload intracellular accumulation and retention for ADCs are critical therapeutic attributes. Although HER2 preferential overexpression on the surface of tumor cells is attractive, its poor internalization and trafficking to lysosomes has been linked to poor therapeutic outcomes. To help address such issues, this review will comprehensively detail the most relevant findings on internalization and cellular accumulation for approved and investigational anti-HER2 antibodies and ADCs. The improved clarity of the HER2 system could improve antibody and ADC designs and approaches for next-generation anti-HER2 and other receptor targeting agents.

## 1. Introduction

We live in the era of the antibody and its advantages and limitations provide incentives for significant increased spending for research activities by the biopharmaceutical (e.g., a biological is the predominant molecular format) sector. The global pharmaceutical market will exceed $1.5 trillion by 2023, growing at a 3–6% compound annual growth rate (CAGR) over the next five years [1]. The biopharmaceutical industry accounts for approximately 17% ($269 billion) of the pharmaceutical market [2]. There is much interest in biopharmaceuticals as they offer several potential benefits including receptor-specific targeting of diseased cells and fewer side effects. Monoclonal antibodies (mAbs) or antibody-based (e.g., immunoglobulin is the predominant molecule) agents dominate biopharmaceutical approvals. Since the creation of the hybridoma technology by Milstein and Kohler in 1975 [3], significant advances have matured this technology to where mAbs can presently and readily be generated to possess exquisite specificity and affinity to a diverse array of proteins [4]. In addition, mAb pharmacokinetic profiles, including engineered versions, are well established. As a result, mAb approvals have increased in first-time total approvals from 27% from 2010 to 2014 to 53% from 2016 to July 2018 [5] (Figure 1a). Not surprisingly, the percentage of overall biopharmaceutical sales is also dominated by mAbs (Figure 1b). With >550 antibody drugs currently in clinical trial [6], it is certain antibodies will accelerate growth for the biopharmaceutical sector and impact the health care system for the foreseeable future.

At a glance, the enormous research development and the increasing availability of biopharmaceuticals would suggest that the experts hit the mark, that these drugs would provide more personalized and effective treatments for disease. Some 40% of the >6000 products currently in clinical development globally are biopharmaceuticals. This suggests that the enthusiasm for biopharmaceuticals will continue into the future. However, private and public health care systems face challenges keeping the cost of biopharmaceuticals low, in particular antibodies and antibody-based drugs. There must be a balance between drug affordability for patients and health care systems that results in good cost-effectiveness, where patients can achieve long-term disease-free survival.

The focus of this review is to address what Carter and Lazar have called the “pursuit of the high-hanging fruit”, specifically, the need for antibody-based drugs to more efficiently access the intracellular space while maintaining target cell specificity [6]. This delivery system is described as the ability of antibodies to bind to target receptors on the surface of diseased cells and efficiently internalize and accumulate inside the cell. Antibody–drug conjugates (ADCs) are the most mature offshoot of antibody therapeutics. ADC therapeutic efficacy is reliant on the ability to efficiently internalize and accumulate the delivered cytotoxic drug inside diseased cells. For unmodified antibodies, increased intracellular access can result in proportional increased target receptor degradation, which, if linked to cytotoxicity, will increase therapeutic efficacy. For the development of next-generation antibodies and ADCs, this review focuses on the lessons learned with the well-studied tumor antigen human epidermal growth factor receptor 2 (HER2) by describing the reports that investigate internalization and accumulation with approved and investigational HER2-specific antibodies and ADCs.

This review examines the relevant evidence and reveals that HER2 does indeed efficiently internalize, albeit under special circumstances. Most importantly, HER2 efficiently internalizes when tumor cells express relatively low levels of HER2. In addition, this review provides a basis to predicatively design and evaluate effective antibodies and ADCs reliant on intracellular access as part of their cytotoxic mechanism of action. This review found that HER2 can be induced to efficiently internalize by (1) crosslinking HER2 receptors, (2) using an antibody that couples HER2 to a known internalizing antibody, and (3) targeting HER2 domain 1. Lastly, this review aims to harmonize future research by providing information on relevant cells lines and associated HER2 expression levels, and on the internalization and cell accumulation methods utilized.

## 2. Antibodies and Antibody-Based Drugs Are Expensive but Are They Cost-Effective?

Antibody drugs are expensive to develop and although they have demonstrated clinical advantages, many have been met with scrutiny, struggling to balance fiscal sustainability, patient access, and innovation [7]. In other words, are antibodies cost-effective? Unmodified mAbs that are reliant on an antagonistic (blockade of receptor–ligand or receptor–receptor interactions) mechanism of action may show some therapeutic potency—the effects tend to be various and ultimately not curative [4]. Yet, their use in patients is unquestioned, as there are >30 antibodies that rank in the top 200 selling prescription medicines, and 5 antibodies in the top 10 [8].

The cost-effectiveness of antibodies and ADCs is debated. Cost-utility analysis (CUA), which is a well-accepted method of evaluating and comparing the relative health outcomes and resource costs associated with different health interventions, enables decision makers to allocate financial resources to maximize health improvements in efficient ways [7]. Neumann and colleagues, who examined published CUAs spanning several therapeutic intervention sectors from 1976 to 2009 or performed head-to-head comparisons, discovered that biopharmaceuticals were less cost-effective, compared with other categories of health interventions, such as conventional pharmaceuticals (e.g., small molecules), immunizations, and diagnostics [7]. In addition, when applied to malignant neoplasms, biopharmaceuticals were also found to be not cost-effective. However, a 2014 study determined that biopharmaceuticals tend to offer greater health improvements over the current standard of care than traditional pharmaceuticals [9]. What is not debated is that biopharmaceuticals are associated with significant elevated costs. Biopharmaceuticals, as ‘specialty drugs’, often have annual costs greater than $50,000 per year and antibodies even greater [7].

What does this specifically signify for antibodies? For example, Cetuximab (Erbitux^®^), an unmodified mAb for metastatic colorectal carcinoma, had poor cost-effective parameters yet costs an average of $80,000 USD to prolong the life of one patient by 1 year [10]. In 2019, Atezolizumab (Tecentriq^®^), the checkpoint inhibitor mAb, was the first of its class approved in the U.S. and Europe for triple-negative breast cancer. However, the United Kingdom’s National Institute for Health and Care Excellence (NICE) rejected Tecentriq, as it was not cost-effective for England’s National Health Service (NHS) [11].

The strategy to conjugate mAbs to agents such as chemotherapy drugs, commonly referred to as ADCs, is one of the most clinically successful offshoots of mAb therapy for oncology. ADCs utilize mAb selectivity to deliver a conjugated cytotoxic agent to tumor cells, while sparing healthy cells. In essence, ADCs are ‘guided missiles’, packing a powerful chemotherapeutic punch specifically against tumors. Mechanistically, ADC effectiveness is reliant on successful tumor cell binding to the cell surface receptor, followed by internalization into the cell and encapsulation inside endosomes. Motor proteins then naturally traffic endosomes to lysosomes for membrane fusion and transfer of the encapsulated contents. Lysosomal proteases digest the antibody backbone or cleave the chemical crosslinker and liberate functional chemotherapeutic metabolites. Depending on the metabolite, the chemotherapeutics can permeate the lysosomal membrane or are presumably transported out of the lysosome by a lysosomal membrane transporter. Several reviews exist on the fundamental mechanisms of how ADCs operate and can be found here [12,13,14].

Although ADCs have converted metastatic/resistant cancer from a death sentence and have improved outcomes, many patients, particularly for solid tumors, develop resistance, and poor cost-effectiveness has been linked to these drugs. Until 2019, there were only four approved ADCs and only one against a solid tumor. There are now eight approved ADCs, with three approved in 2019 and one as recently as January 2020. In 2013, trastuzumab-emtansine (T-DM1; Roche) was approved for use for patients with overexpression of HER2-positive, unresectable locally advanced or metastatic breast cancer who previously had been treated with trastuzumab and a taxane (paclitaxel or docetaxel) but whose disease had progressed [15]. The recommended treatment options for patients with HER2-positive breast cancer are: trastuzumab in combination with paclitaxel (first-line); capecitabine (including in combination with lapatinib) or vinorelbine (including trastuzumab when there is evidence of progression in the central nervous system) (second-line); and vinorelbine or capecitabine or trastuzumab (third-line) [16]. The mAb pertuzumab in combination with trastuzumab and chemotherapy is also recommended as a first-line treatment option (this is discussed more in Section 5.2A) [17]. T-DM1 approval was based on the EMILIA study that demonstrated that T-DM1 extended median progression-free (PFS) and overall survival (OS) of patients with HER2-positive metastatic breast cancer for 9.6 months and 30.9 months, respectively, as a single agent as compared with standard combination therapy (PFS = 6.4 months and OS = 25.1 months), and was associated with fewer adverse events [18]. The TH3RESA trial, which evaluated T-DM1 versus physician’s treatment choice supported the findings from the EMILIA trial [19]. Hence, in the metastatic setting, the life extension T-DM1 provided patients was approximately 6 months to 1 year post treatment initiation. Roche recommended T-DM1 as a second-line option. In 2019, T-DM1 was additionally approved as an adjuvant treatment for HER2-positive early breast cancer when the patient had taken neoadjuvant trastuzumab and chemotherapy and there was still cancer remaining [20]. Approval was based on the KATHERINE trial, with randomized patients to receive T-DM1 or trastuzumab. At three years, the proportion of patients that remained event free for invasive disease in the T-DM1 arm was 88.3% compared to 77.0% for trastuzumab [20]. Although these are clear demonstrations of the ability of T-DM1 to improve patient outcomes when administered as first-, second-, and third-line treatments, T-DM1 costs approximately $9800 per month, or $94,000 for one treatment course, and the price can rise to >$124,000 for patients weighing >100 kg [15,21].

In 2014, England’s NICE, which routinely conducts health economic evaluations for new drugs costing more than £30,000, determined that T-DM1 did not produce the quality-adjusted life years (QALY) gains required to justify it price [15]. QALY is a measure of the incremental health improvement provided by a new treatment compared to previous treatment options. NICE’s Evidence Review Group calculated the cost to be in the region of £167,000/QALY gained. In comparison, lapatinib plus capecitabine cost £49,798/QALY gained. The Lancet published an editorial disapproving of NICE’s decision, alleging that it did not include clinical ‘value’, which is what counts the most [22]. However, NICE did not base its decision solely on cost. NICE justified its decision based on a ‘de novo cohort state transition’ model, as there was no economic evaluation of T-DM1 in existence at the time [23]. The ‘model’ had three health states—PFS, progressed disease, and death—and uses weekly time periods for assessment [15]. The EMILIA trial data as well as T-DM1 production costs were inputted into the model and determined that T-DM1 would not be considered a cost-effective use of NHS resources.

The NHS has a dedicated Cancer Drugs Fund with a set budget, and NICE has stated that “the budget for health technologies is likely to be allocated more inefficiently as relatively expensive cancer drugs are being funded in preference to drugs which represent a more cost-effective use of public funds with other disease areas”. This automatically may place NICE in a situation of ‘negative’ guidance for expensive anti-cancer drugs when clinical benefit is not remarkable, such as the case with T-DM1. Nonetheless, NICE’s decision has been validated by health economic analyses from academia and other governmental agencies. Le et al. examined the cost-effectiveness of T-DM1 in the US health care system and also used models that focused on clinical efficacy [24]. The author’s determined that T-DM1 costs up to $220,385/QALY. Since the study calculated a willingness-to-pay threshold of $150,000/QALY, this also showed that T-DM1 was not cost-effective when analyzed through the lens of the US health care system and, hence, supports the decision by NICE. Although in mid-2017 a confidential agreement was struck (terms not disclosed) between Roche and the NHS that made the drug widely available again [25], the findings by Le et al. underscore the dilemma with ADCs [24], which have been further supported by governmental health agencies in Canada, Australia, and Spain [26,27].

In December 2019, trastuzumab-deruxtecan (T-DXT; Enhertu^®^; AstraZeneca/Daiichi Sankyo) was approved for use with patients with metastatic breast cancer who have received two or more prior anti-HER2 therapies [28]. In the phase II DESTINY-Breast01 trial, T-DXT was evaluated in patients who had undergone a median number of six lines (range 2 to 27) including T-DM1 [28]. The study was multicenter and included 184 patients that were administered three different doses to establish a recommended dose and, then, efficacy and safety of the recommended dose was evaluated. The trial showed an overall response rate of 60.9%. The response was based on the modified Response Evaluation Criteria in Solid Tumors, which assesses the change in tumor burden. Patients had a PFS of 14.8 months. Notably, 13.6% of patients developed lung damage, of which 2.3% was of grade 5. T-DXT is now in phase III trial but based on the results of the phase II trial, the US Food and Drug Administration granted an accelerated approval. Based on the speed of approval, it is highly unlikely that a comprehensive health economic evaluation has been performed. Nonetheless, T-DXT will launch in 2020 and health care investment banking specialists have priced it at a per-patient cost of approximately $13,300 per month [29]. T-DXT piques curiosity as it is a breakout treatment and only time and additional findings will fully determine its potential and limitations and, ultimately, whether it is cost-effective. NICE has scheduled the appraisal of the clinical and cost-effectiveness of T-DXT and their finding is anticipated later in 2020 [30].

## 3. The HER2 System for Describing the Challenges of Antibodies Accessing the Intracellular Space

HER2 belongs to the family of ErbB receptor tyrosine kinases that includes epidermal growth factor receptor (EGFR)/ErbB1, HER3/ErbB3, and HER4/ErbB4. EGFR, HER3, and HER4 bind several different ligands, and no ligand has yet to be identified for HER2 [31]. HER2 activity is subsequent to homo- and heterodimerization with other family members—it promotes cell growth and division when expressed at normal levels (≤45,000 receptors/cell) [32]. When overexpressed (up to 2 × 10^6^ receptors/cell), HER2 promotes cell growth beyond normal limits and this is associated with an aggressive breast cancer phenotype and a poor prognosis [33,34,35]. HER2 is overexpressed in 15–20% of breast tumors [36]. In addition, HER2 overexpression is now linked to a number of diverse cancers and tumor progression and is reviewed in [37].

One of the great examples of antibody-targeted therapy for cancer is the story of HER2. The mAb trastuzumab therapy became available approximately 20 years ago. Trastuzumab in combination with paclitaxel has clearly benefitted patients. The preferential overexpression of HER2 on the surface of tumor cells makes it an excellent choice to target with mAbs. There are several mechanisms of action of trastuzumab and they are reviewed in [38]. One important mechanism is thought to be the binding of HER2 followed by internalization and subsequent trafficking to lysosomes and degradation and, hence the reduction in aberrant signaling. However, trastuzumab is ultimately not curative [38]. In regards to T-DM1, therapeutic effectiveness is reliant on successful tumor cell binding to HER2, followed by sufficient internalization into the cells and efficient trafficking and degradation in lysosomes. For ADCs in general, the parameters that affect receptor internalization, and subsequent retention in cells are vital to the intracellular access efficiency required for effective cytotoxicity and tumor killing. The aim of an ADC is to increase the therapeutic index by specifically delivering the chemotherapeutic to the tumor and also decrease systemic toxicity. However, an ideal delivery system must enable ADCs to efficiently access the intracellular environment and be degraded, enabling release of the transported cytotoxin to accumulate to sufficient concentrations in order to evoke effective tumor cell killing.

Unfortunately, most breast cancer patients treated with T-DM1 acquire resistance [39]. In addition, T-DM1 failed to improve patient outcome in HER2-positive gastric cancer [40]. The implicated resistance factors include increased expression of ATP-binding cassette transporters [41,42] and HER2 reprocessing after receptor-mediated internalization that suggests either incomplete trafficking of T-DM1 to lysosomes [43] and/or rapid recycling back to the plasma membrane [44,45]. These resistance mechanisms, each on their own or cumulatively, ultimately result in T-DM1 being unable to accumulate DM1 sufficiently to evoke effective tumor cell death. For next-generation unmodified antibodies, ADCs, or other receptor-targeted therapeutics that utilize intracellular access exclusively or as part of their cytotoxic mechanism of action, an essential question is how important and how much intracellular access and/or accumulation of the delivered payload is required?

Bertelesen and Stang have previously declared HER2 trafficking as “mysterious” [31]. Although HER2 internalization is debated, it is mostly accepted that HER2 has no or limited internalization and subsequent trafficking to lysosomes and trastuzumab binding does not increase internalization [44,46]. This review details the relevant evidence and, more importantly, aims to discover opportunities for effective next-generation antibody-based therapeutics. As the evidence reveal, apart from the preferential overexpression of HER2 on tumor cells relative to healthy tissues, HER2 is not an ideal target, yet trastuzumab and T-DM1 have made tremendous clinical impacts. This makes HER2, in the context of antibody targeting, an alluring receptor to study.

### 3.1. Antibody HER2 Has Limited Internalization

Studies from the mid-2000s established the notion that HER2 has poor internalization. This finding, although challenged in the next section, continues to have support from recent studies that also show poor HER2 internalization. It is important to note that in SKBR3 breast cancer cells that express approximately 2 × 10^6^ receptors/cell and are considered high (Table 1), trastuzumab induced internalization and subsequent intracellular trafficking resembles non-bound HER2 [44,46]. Hence, trastuzumab binding does not induce any increased internalization. Austin et al. and Longva et al. originally evaluated radioiodinated (^125^I) trastuzumab and measured cell surface and intracellular radioactivity at various time points out to 24 h in SKBR3 cells [44,47]. At best, the fraction of ^125^I-trastuzumab that internalized was at least 40-fold less than the fraction remaining at the cell surface. There was also poor internalization in MCF7 cells that are considered to have low HER2 expression levels. Austin et al., in the same study, reported that any internalized HER2/trastuzumab was rapidly recycled to the cell surface [44]. More recent studies have corroborated these earlier reports by demonstrating that trastuzumab did not internalize in HER2 high and intermediate, expressing BT474 and OE19 cells, respectively [48,49]. Hence, there is strong evidence that indeed HER2 has poor internalization and trafficking to lysosomes.

It is well known that HER2 is the preferred dimerization partner in the HER family and these heterodimers have a decreased ability to undergo internalization [50]. Valabrega et al. showed that trastuzumab internalization in HER2 high SKBR3 cells was inhibited when in the presence of the EGFR ligand tumor growth factor α (TGFα) [51]. It is known that EGFR interacts with several ligands that induce internalization [50]. This study demonstrated that trastuzumab was not internalized because TGFα binding induced EGFR into an active conformation that easily formed heterodimers due to the increased presence of HER2. As a result, this reduced the available HER2 interaction sites for trastuzumab and, hence, limited the amount of internalized antibodies.

### 3.2. Antibody HER2 Is Efficiently Internalized and Is Processed in Lysosomes

However, other studies have demonstrated that trastuzumab does efficiently internalize but is based on key requirements. Ram et al., using fluorescently-conjugated trastuzumab, showed that it could internalize in a panel of breast and prostate cancer cell lines (Table 1) [52]. Interestingly, trastuzumab internalization efficiency was inversely proportional to HER2 expression levels. Nearly 75% of trastuzumab internalized in cells with HER2 expression levels up to 33-fold lower than high HER2-expressing cells. However, trastuzumab recycling was proportional to the level of HER2 expression. Across the entire cell line panel with various HER2 expression levels, trastuzumab cell surface levels decreased by ~15% within the first 60 min. By 18 h, the original cell surface levels of HER2 reappear in HER2 high expressing cells. In contrast, HER2 cell surface levels continued to decrease to ~15% and ~50% of the original levels in the HER2 low and intermediate expressing cells, respectively [52]. For lysosomal processing, trastuzumab was localized with Lamp-1 (lysosomal marker) in all cells with HER2 expression. However, there was stark contrast in the distribution within the intracellular vesicles of interest. Specifically, in high HER2-expressing SKBR3 cells, trastuzumab was localized to an area known as the ‘limiting membrane’ of transferrin receptor-positive endosomes. In contrast, in low HER2-expressing LAPC-4 cells, trastuzumab was localized in the lumen of transferrin receptor-positive endosomes. The latter is known to enter the late endosomal/lysosomal pathway [53]. The former instead routes back to the cell surface. The molecular mechanism explaining why cells with low HER2 expression traffic to lysosomes as opposed to recycling back to the cell surface remains a mystery and requires further studies. Nonetheless, the study showed that trastuzumab/HER2 could efficiently internalize in a reduced HER2 expression environment and that expression and not cell of origin was the principle determinant of HER2 internalization efficiency.

There is additional strong evidence that supports and also contradicts the findings by Ram et al. that internalization efficiently occurs in cells with low levels of HER2 expression and offers the following molecular and cellular reasons:Diessner et al. showed that trastuzumab could internalize in HER2 low expressing cells [54]. Primary tumor cells from metastatic breast cancer patients were cell sorted into a putative cancer stem cell (CSC) population with a CD44 high/CD24 low/HER2 low phenotype [55]. Trastuzumab, directly labeled with a pH-sensitive dye, showed that >50% of the HER2-bound antibody internalized into acidic intracellular vesicles in these HER2 low breast CSCs. In contrast, <2% of trastuzumab internalized in CD44 high/CD24 high/HER2 high non-CSCs. The increased internalization in CSCs was associated with improved cytotoxicity when treated with T-DM1. In the same study, HER2 low expressing cell lines such as MCF7 were also susceptible to T-DM1. The enhanced trastuzumab and T-DM1 internalization was linked to autophagy, a regulated catabolic process that involves the degradation of a cell’s own components via the lysosome [56]. Because autophagy is known to recycle cellular components and, thus, enabling maintenance of cells, especially stem cells, the authors proposed that low HER2 expression was mainly due to internalization via autophagy. This work also slightly contradicts Ram et al., as the results suggested that cell origin is relevant for HER2 internalization.De Goeij et al. studied a novel panel of anti-HER2 mAbs for internalization in several HER2-positive cell lines [57]. It was revealed that, especially in HER2 low A431 and Colo205 cells, antibodies that did not interfere with HER2 heterodimerization induced effective internalization. In contrast to the results from Valabrega et al. (Section 3.1), this study suggested that the formation of HER2/ErbB heterodimers enhances antibody internalization and, importantly, heterodimer formation was more frequent in cells with low HER2 expression.Fehling-Kaschek et al. revealed that internalization did not occur in SKBR3 cells when HER2 was located in areas where the membrane was smooth [58]. When located in areas where the membrane was ruffled, trastuzumab was able to efficiently internalize. Interestingly, there were higher HER2 densities on membrane ruffles than in the flat regions. It is unclear whether the membrane topology is ruffled or smooth in additional cell types that express low levels of HER2. This most likely indicates that membrane ruffling activity may be an important aspect for internalization. Membrane ruffling is known to occur at distinct cellular zones undergoing rapid reorganization of the plasma membrane [59]. Interestingly, membrane ruffling was observed in EGFR-positive cells when stimulated with EGF [60]. In addition, membrane ruffling has been associated with cancer cell migration and invasion [61,62].

Cumulatively, these studies indicate that HER2 and bound antibodies do indeed internalize most likely more efficiently when expressed at low (≤1 × 10^5^ rec/cell) relative to high (≥1 × 10^6^ rec/cell) levels but conditions such as membrane ruffling can enable internalization at areas of the cell with localized HER2 high densities.

### 3.3. How Does the HER2 Internalization Profile Impact ADCs?

T-DM1 efficacy is dependent on the level of expression of HER2 on cancer cells and it is known that patients who express HER2, defined by immunohistochemistry 3+, have more frequent responses than patients with reduced levels [85,86]. Despite the findings in the previous section that trastuzumab/anti-HER2 mAbs have preferential and efficient internalization for cells with low HER2 expression, parallel studies during the development of what resulted in T-DM1 demonstrated that various trastuzumab-conjugate versions were highly potent in HER2 high expressing cells. In addition, Lewis Phillips et al. showed that T-DM1 was not cytotoxic against MCF7 cells while Erickson et al. showed that T-DM1 was highly potent against MCF7 cells [69,87]. This indicates that there must be, at least, a sufficient fraction of trastuzumab ADCs that is processed in the lysosome for DM1 release. For cells with HER2 high expression, poor internalization may be compensated by the excess amounts of HER2 at the cell surface that enough receptor is still internalized. For cells with HER2 low expression, the efficient internalization may compensate for reduced amounts of HER2 at the cell surface. In addition, the differences in methodologies between internalization and cytotoxicity assays can provide a justifiable explanation. The antibody internalization studies were performed in a time frame ranging from minutes to up to 72 h. In contrast, cytotoxicity studies, in general, are evaluated at or beyond 72 h. To compare ‘apples to apples’, anti-HER2 ADC internalization studies are detailed.

Studies using confocal microscopy, similar to that performed with unmodified anti-HER2 antibodies indeed, revealed that T-DM1 and other trastuzumab-based ADCs are internalized and processed inside lysosomes. Cilliers et al. modified T-DM1 with a fluorescent dye that is hydrophilic and residualizing (trapped inside cells) and lipophilic and non-residualizing (washes out) [88]. As T-DM1 binds to HER2 at the cell surface, if it is internalized and localized to lysosomes and subsequently degraded, the lipophilic dye will diffuse out of the cell, while the hydrophilic dye will remain trapped. The fluorescence signal of each dye can be measured and the ratio calculated to determine the lysosomal processing efficiency. Treatment of N87 and HCC1954 cells, and analysis at 12 h and 24 h, revealed that T-DM1 degradation did occur in lysosomes and the signal for the lipophilic dye decreased and the signal for the hydrophilic dye was retained inside punctate vacuoles identified as lysosomes. Although both cell types express HER2 at high levels, the HCC1954 cell line degraded T-DM1 faster than N87 and was reasoned for why HCC1954 was more sensitive to T-DM1 relative to N87. Wang et al. determined that approximately 10% of bound T-DM1 internalized in N87 cells and much of it was recycled back to the cell surface [82]. Nonetheless, T-DM1 was still cytotoxically potent and had anti-tumor activity. Kulkarni et al. studied the lysosomal processing of trastuzumab conjugated to the cytotoxin thailanstatin, a compound that inhibits function of the spliceosome [89]. Thailanstatin was modified with a fluorophore to remain with the payload, while the trastuzumab surface lysines were modified with a separate fluorophore and allowed for simultaneous intracellular tracking of the drug and antibody components. Time-course trafficking studies up to 24 h in SKOV3 cells showed that both drug and trastuzumab colocalized inside lysosomes efficiently and recycling was not indicated. At 24 h, fluorescence signal inside lysosomes overpowered any visible fluorescence originating from the cell surface indicating efficient internalization and subsequent trafficking to lysosomes. Thus, internalization studies with trastuzumab-based ADCs or T-DM1 demonstrate that at least some internalization does occur that is sufficient enough for potent cytotoxicity. This suggests the possibility that even cells with high HER2 expression that should have poor internalization can in fact have sufficient internalization. However, the relationship between HER2 expression and kinetic internalization and lysosomal processing was not explored in HER2 low expressing cells. Thus, molecular reasons for why anti-HER2 ADCs against MCF7 were cytotoxic remain a mystery.

Molecular evidence of intracellular processing exists and supports anti-HER2 ADC internalization and localization to lysosomes. Hamblett et al. demonstrated that the lysosomal membrane protein SLC46A3 was responsible for transporting the lysine-SMCC-DM1 metabolite from inside lysosomes to the intracellular space in SKBR3 cells and was required for cytotoxicity [90]. Wang et al. demonstrated that the level of microtubule disruption was proportional to increasing concentrations of T-DM1 in BT474 cells [91]. Caculitan et al. demonstrated that cysteine cathepsin proteases in the lysosome are part of the mechanism involving trastuzumab ADC efficacy in BT474 and KPL4 cells [74]. Baldassarre et al. demonstrated efficient internalization in HER2 high expressing HCC1954 and SKBR3 cells [64]. When the endocytic scaffolding protein endophilin A2 was knocked down, T-DM1 was ineffective at cell killing.

There is convincing evidence supporting internalization and lysosomal processing for T-DM1/trastuzumab-based ADCs. More importantly, the level of internalization appears to be sufficient to cause potent cytotoxicity. However, the above studies did not include cells with low HER2 expression levels. Having a comparison of high versus low levels of HER2 expression would have allowed for elucidation for relative internalization efficiency. This could have further been associated with the relative levels of cytotoxic potency. As previously described, Diessner et al. showed that T-DM1 was more effective in CSCs with low HER2 expression [54]. Diessner et al. also showed that T-DM1 was effective at killing MCF7 HER2 low cells and supports the findings of Erickson et al. [54,69]. Thus, the above internalization data does not answer why, in general, T-DM1/anti-HER2 ADC cytotoxicity decreases proportionally in cell lines with reduced HER2 expression, when the evidence points to poor internalization in cells with high HER2 expression and robust internalization in cells with low HER2 expression. Cilliers et al. eluded to differences in the kinetic processing of T-DM1 between different cell lines. Rather than classifying internalization and lysosomal processing as being poor or limited, it is just kinetically slow. The slow processes may still allow the ADC to gradually accumulate the cytotoxic payload in cells over time to levels sufficient to result in potent cytotoxicity. This could be a justifiable explanation for cells that express high levels of HER2 but does not explain why T-DM1 and anti-HER2 ADCs in general are less effective in HER2 low expressing cells. More studies will have to be conducted because if increased internalization efficiency occurs in cells with low HER2 levels relative to high levels, T-DM1 should also be effective in patients with 1+ and 2+ HER2 expression.

## 4. ADC Drug Cellular Accumulation and Impact on Cytotoxicity and Tumor Killing

Having insufficient evidence to fully grasp the relationship between HER2 internalization, sufficient ADC intracellular access, and cytotoxic impact, this section presents studies that have evaluated and quantified cellular payload accumulation levels and linked it to cytotoxic impact. We take note of cell types and the corresponding HER2 expression levels, which can be found in Table 1. ADC processing has been studied in great detail in the past by a variety of experimental methods. A total of 35 publications were identified via PubMed and Google Scholar using the key words: ‘antibody, conjugate, accumulation, cellular, intracellular, and tumor’. Selected studies had to contain quantitative information on the cellular concentration of the delivered cytotoxic drugs. Studies with qualitative cellular accumulation results were screened out. The internalization studies previously described were mostly based on fluorescence-based approaches that can quantify relative internalization and areas of antibody intracellular localization. However, liquid chromatography–mass spectrometry/mass spectrometry (LC–MS/MS) methods are able to quantify absolute cell-associated drug concentrations due to calibration curves that can be used to precisely measure ADC metabolite concentration in given samples. With the exception of one study that used radiolabeled-tagged ADCs results converted into absolute quantities, we evaluate the findings of cellular payload accumulation using LC–MS/MS. However, the LC–MS/MS methods still vary considerably in number of cells, ADC concentration used, incubation time point, and other aspects, thus the results have to be taken with a certain level of caution. The different methods are listed in Table 2. In addition, all studies that quantified payload cellular accumulation in vitro were trastuzumab-based ADCs or T-DM1.

Erickson et al. was the first to evaluate cellular payload accumulation for trastuzumab-based ADCs [69]. Trastuzmab was conjugated to DM1 using the non-cleavable linker *N*-succinimidyl 4-(N-maleimidomethyl)cyclohexane-1 carboxylate (SMCC) or N-succinimidyl 4-(2-pyridyldithio)pentanoatel (SPP) (Figure 2). The trastuzumab-sensitive breast cancer cell lines SKBR3 and BT474 were all equally sensitive to T-DM1 and T-SPP-DM1. However, T-DM1 had significantly increased cytotoxicity relative to T-SPP-DM1 against the trastuzumab-insensitive breast cancer cell lines BT474EEI and MCF7. Interestingly, both T-DM1 and T-SPP-DM1 killed BT474EEI tumors with equal potency when administered in equivalent doses. Characterization of the accumulated DM1 in BT474EEI cells revealed that the major DM1 catabolites reached maximum values of 300 and 200 nmol/L for T-DM1 and T-SPP-DM1, respectively, at 24 h (Table 3). The increased cellular accumulation of DM1 was caused by a 30% increase in T-DM1 bound to the cell surface after several wash steps. In other cell lines, the only data reported was that T-DM1 and T-SPP-DM1 accumulated DM1 metabolites in MCF7 cells at levels of 500 and 150 nmol/L, respectively. The increased cellular accumulation of DM1 by T-DM1 in MCF7 cells supports the increased internalization and lysosomal processing for low HER2-expressing cells but also contradicts that T-DM1 is ineffective in HER2 low expressing cells, as described in the previous section. T-DM1 accumulated DM1 at 500 nmol/L in SKBR3 cells. The amount of DM1 accumulation for T-SPP-DM1 in SKBR3 cells was not clearly determined.

For tumor accumulation, DM1 metabolite concentrations in BT474EEI xenografts were ~700 and 500 nmol/L for T-DM1 and T-SPP-DM1, respectively. Pharmacokinetic studies revealed that T-DM1 had a ~2-fold slower plasma clearance than T-SPP-DM1. This was likely due to the greater stability of the SMCC linker. The increased plasma retention resulted in increased tumor exposure for T-DM1 relative to T-SPP-DM1. However, this did not result in increased tumor killing by T-DM1. This study indicated two major findings. One, T-SPP-DM1 could be effective due to a ‘bystander effect’ reviewed in [93]. Two, T-DM1 did not internalize and/or become degraded efficiently to evoke increased cytotoxicity and tumor killing in the BT474EEI xenograft model. This is further supported by T-DM1 being found retained on the surface of BT474EEI cells more than T-SPP-DM1.

Caculitan et al. studied the relationship in payload cellular accumulation in trastuzumab-monomethyl auristatin E (MMAE) that were modified with a protease-sensitive linker (Figure 2) and in combination in KPL4 cells that had cathepsin B knocked down [74]. MMAE is a payload on the approved ADCs brentuximab vedotin (Adcetris^®^) and enfortumab vedotin (Padcev^®^). These ADCs utilize a ‘traceless’ linker containing valine-citrulline (vc) that is a substrate for the cysteine protease cathepsin B [94]. The vc-linker joining the antibody to MMAE contains the *p*-aminobenclyoxycarbonyl self-immolative group that upon cleavage results in a ‘clean’ release of the drug. The study found no statistically significant difference in the cellular-associated free MMAE concentrations in KPL4 parental and cathepsin B knocked down cells at all concentrations evaluated at 24 h. Unfortunately, there were no studies evaluating HER2 internalization in KPL4 cells, so internalization efficiency is unknown. The HER2 expression levels have been estimated as intermediate or high based on the gene amplification study [73]. A vc linker steroisomer was synthesized that was resistant (R) to cathepsin B. The T-vc(R)-MMAE was still cytotoxically potent, albeit attenuated, against cathepsin B knocked down cells compared to T-vc-MMAE. T-vc-MMAE and T-vc(R)-MMAE accumulated ~450 and ~125 nmol/L MMAE catabolites in KPL4 cell, respectively. T-vc-MMAE and T-vc(R)-MMAE had IC_50_ values of 0.034 and 0.085 μg/mL, respectively. Tumor killing studies were not performed. Nonetheless, this study demonstrated that the 3.6-fold increase in cellular accumulation with the protease-sensitive linker-containing ADC did roughly correspond to a proportional increase in cytotoxic activity relative to the protease-resistant linker-containing ADC.

Sauveur et al. provided insight on T-DM1 cellular accumulation and associated insights in esophageal cancer [79]. The OE19 cell line was studied along with two clones resistant to T-DM1. OE19TCR and OE19TR were generated by continuous increased exposure to T-DM1 in the presence or absence of ciclosporin A, respectively. Ciclosporin A is an inhibitor of the drug efflux transporter MDR1. Resistance to ADCs transporting maytansinoid payloads has been reported to be mediated by MDR1 [95]. The OE19-resistant cells contained increased MDR1 levels relative to OE19-sensitive cells. Importantly, HER2 expression was not affected in the resistant cells or its ability to bind T-DM1. There was slightly more DM1 accumulated in OE19TR cells relative to the parental OE19 cells. Yet, the T-DM1 IC_50_ value was nearly 15-fold higher in OE19TR cells. In the OE19TCR cells, DM1 accumulation was ~40% lower than in the parental OE19 cells. Accordingly, T-DM1 had increased difficulty to kill OE19TCR cells with an IC_50_ value almost 20-fold higher compared to parental OE19 cells. Tumor killing studies were not performed. Nonetheless, this study showed the complexity with T-DM1 resistance, where one model had decreased cellular accumulation, relative to sensitive cells, and increased resistance caused by the increase in MDR1 expression. Another T-DM1 resistance model had increased cellular accumulation, relative to sensitive cells, but maintained resistance.

In gastric cancer, Wang et al. showed that T-DM1 accumulated significantly less DM1 metabolites in T-DM1-resistant N87KR cell lines relative to the parental N87 cells [82]. The N87KR cells expressed the same levels of HER2 as the parental cell line but had aberrant vacuolar H^+^-ATPase (V-ATPase) activity, which resulted in attenuated lysosomal acidification. As a result, T-DM1 accumulated 1 ng/200,000 (4.9 nmol/L) and 0.65 ng/200,000 cells (3.2 nmol/L) of DM1 metabolites in N87 and N87KR cells, respectively. The slight increase in DM1 metabolite accumulation in N87 over N87KR cells resulted in an increase in the T-DM1 IC_50_ value by a factor of 65. Interestingly, the study determined that only ~10% of T-DM1 was internalized in both cell lines. VATPase activity was confirmed by treating sensitive N87 cells simultaneously with the selective VATPase inhibitor bafilomycin that caused a reduction in DM1 metabolite cellular accumulation. Altered lysosomal processing has previously been reported as a T-DM1 resistance factor in N87 cells [43]. Resistance was overcome when a different anti-HER2-vc-MMAE ADC was tested. Anti-HER2-vc-MMAE cytotoxicity assays determined equivalent IC_50_ values (~60 nmol/L) for N87 and N87KR cells, albeit at significantly reduced potency relative to the T-DM1/sensitive N87 system. Unfortunately, the cellular accumulation MMAE levels were not determined. In vivo, both T-DM1 and anti-HER2-vc-MMAE appeared to be effective at killing N87 xenografts and, anti-HER2-vc-MMAE killed N87KR xenografts better than T-DM1 but extensive comparative evaluations were not performed. This study linked lysosomal processing to T-DM1 resistance and cellular payload accumulation.

From a safety perspective, Uppal et al. discovered that very low amounts of accumulated DM1 metabolites was sufficient to effectively kill the hematopoietic cells, megakaryocytes [92]. Thrombocytopenia (low blood platelet count) has been shown to be the dose-limiting toxicity with T-DM1 and grade ≥3 thrombocytopenia was observed in 4.7% to 12.9% of patients in phase III studies [18,19,96]. Other hematologic lineages appear to be relatively unaffected by T-DM1 having low rates of anemia, leukopenia, and neutropenia [18,97]. Hematopoietic stem cells (HSCs) were differentiated into megakaryocytes. T-DM1 internalization was dependent on the cellular differentiation stage. The greatest amount of T-DM1 internalization occurred in HSCs and premature megakaryocytes. It was determined that the levels of accumulated DM1 metabolites were up to 0.85, 8.18, and 3.45 nmol/L in HSCs, immature, and mature megakaryocytes, respectively. T-DM1 affected the differentiation of HSCs to immature megakaryocytes having an IC_50_ value of ~50 nmol/L. T-DM1 also affected the differentiation of immature to mature megakaryocytes having an IC_50_ value of ~20 nmol/L. Mature megakaryocytes appeared to be less sensitive. Thus, the level of intracellular accumulation was the highest in the most sensitive differentiation cell type, immature megakaryocytes. Interestingly, control ADCs had equally cytotoxic potency. It was discovered that the immature megakaryocytes lacked HER2 expression and internalization was mediated through antibody Fc binding to FcγRIIa. Nonetheless, this study determined that low cellular concentrations of DM1 metabolites were sufficient to kill important cells in the hematopoietic system.

Although the above studies increased the complexity of HER2 internalization as it relates to cytotoxic effectiveness, in general, these studies indicate that increased cellular payload accumulation results in increased cytotoxic potency. However, there are still several unknowns.

Linker type: There are contrasting findings between Caculitan et al. and Erickson et al. Both the T-SPP-DM1 and T-VC-MMAE result in a ‘traceless’ release of the drug. However, Erickson et al. showed that T-DM1 accumulated 1.5- and 3.3-fold more DM1 metabolites in BT474EEI and MCF7 cells, respectively, relative to T-SPP-DM1. Accordingly, T-DM1 had an increased cytotoxic potency by 5- and 3.5-fold over T-SPP-DM1 in BT474EEI and MCF7 cells, respectively. This demonstrated that an ADC with a non-cleavable linker accumulated better than a reducible linker. However, Caculitan et al. showed the opposite. The T-vc-MMAE accumulated 3.8-fold more drug in KPL4 cells than T-vc(R)-MMAE. Accordingly, T-vc-MMAE had an increased cytotoxic potency by a factor of 2.1.HER2 expression: It is unclear why T-DM1 accumulates the same level of DM1 metabolites in MCF7 HER2 Low as SKBR3 HER2 high cells.Resistant cells: Sauveur et al. demonstrated that actually the resistant OE19TR cell line accumulated more DM1 metabolites (albeit only a 28% increase) than the parental OE19 cells. However, T-DM1 still killed OE19 cells with an increased potency by a factor of 15 over the resistant OE19TR cells. The resistant OE19TCR cells accumulated 39% less DM1 metabolites than in parental OE19 cells. Accordingly, T-DM1 had an increased cytotoxic potency in OE19 cells by 19.6-fold over the OE19TCR cells.All the studies that investigated cellular payload accumulation were trastuzumab-based ADCs or T-DM1. There is no information on other anti-HER2 ADCs.

Unfortunately, the relationship between HER2 expression and cellular accumulation and cytotoxicity remains ambiguous. Due to the limited number of studies that quantified cellular payload accumulation by LC–MS/MS and because each study did not explicitly evaluate accumulation based on different levels of HER2 expression, the conclusions are limited. MCF7 as the only HER2 low-expressing cell line, DM1 metabolite cellular accumulation was compared against the other cell lines with high or intermediate HER2 expression. There was no association for increased cellular accumulation and cytotoxicity for the cell lines SKBR3, OE19, N87, and BT474 relative to MCF7. In contrast, there were increased or equivalent DM1 metabolite accumulation levels in MCF7 relative to the high and intermediate HER2-expressing cell lines. In line with the evidence that HER2 low cells internalize better, T-DM1 was significantly more cytotoxic against MCF7 cells relative to OE19, N87, and BT474EEI. Only SKBR3 was more sensitive to T-DM1.

It is not clear why the HER2 low-expressing MCF7 accumulates more DM1 metabolites than the other HER2 high/intermediate expressing cells. Most likely, differences in quantitative methods and analyses account for the large variations in the reported accumulation levels. In addition, accumulation and cytotoxicity assays are very different in the number of cells, ADC concentration used, and the sample period. For example, in vitro cytotoxicity of ADCs is typically measured at or after three days of continuous incubation with cells. This allows for sufficient time for killing of cells during cellular growth. The methods to determine intracellular drug levels are designed for intact live cells. Thus the main period of time addressed in accumulating studies were ≤24 h post incubation. In addition, the quantitative values for the drug catabolites do not reflect the actual intracellular accumulation levels. Typically, cells are spun down after ADC treatment, washed and lysed. Hence, ADCs bound at the cell surface are incorporated into the evaluation.

Lastly, there is insufficient knowledge about what the correlation between payload tumor accumulation with respect to varying levels of HER2 expression. Erickson et al. did determine that increased DM1 metabolite tumor accumulation was proportional to cellular accumulation in the BT474EEI model. However, no further evaluation was performed with the SKBR3 or MCF7 tumors. Zhang et al. evaluated anti-HER2 antibodies conjugated to pyrrolo[2,1-c][1,4]benzodiazepine-dimers (PBD) in mice bearing Fo5 mouse mammary tumors that stably express human HER2 [98]. The team showed that anti-HER2-PBD improved tumor killing with increasing PBD catabolite amounts in the tumor. They calculated that 1/10^6^ BPD/DNA base-pair was sufficient to achieve tumor growth stasis. Importantly, the authors indicated that increasing the tumor PBD concentration did not improve tumor killing once a PBD concentration ‘threshold’ has been reached. In fact, the authors surmise that increasing the tumor concentration beyond the threshold could contribute to systemic toxicity. However, more in vivo studies are needed to determine the association between cellular and tumor accumulation and in tumors with varying HER2 expression. In summary, the evidence thus far indicates that a necessary amount of payload is required to accumulate in cells and tumors. Too few, and the tumor cell survives. Too much, and there is a risk of causing unwanted toxicity (Figure 3).

## 5. Approaches to Improve HER2-Specific Antibody Internalization and Accumulation

### 5.1. Combination Treatment with Molecular Inhibitors

The small-molecule geldanamycin (GA) has provided a great deal of knowledge and an effective approach to improve trastuzumab and HER2 internalization and lysosomal processing. GA is a benzoquinoid ansamycin antibiotic that inhibits the chaperone Hsp90 [99]. It was previously shown that GA induced HER2 cleavage, which resulted in internalization into endosomes [100]. Austin et al. first linked the importance of HER2 internalization and intracellular trafficking and its impact on the fate of the intracellular distribution of trastuzmab when GA was added to the cell culture [44]. Treatment of SKBR3 cells with GA resulted in a rapid and efficient downregulation of HER2, which was due to reduced recycling rather than increased internalization. As a result, trastuzumab and anti-HER2 ADCs were efficiently routed to lysosomes and degraded [44,101]. The use of GA to improve trastuzumab internalization and lysosomal processing has now been well supported [47,65]. GA and other HSP90 inhibitors in combination with trastuzumab are now in the clinic and they are reviewed in [102,103].

Increasing evidence indicates that there is aberrant metabolism of choline phospholipids in cancer [104]. Specifically, phosphatidylcholine-specific phospholipase C (PC-PLC), an enzyme responsible for phophatidlycholine hydrolysis, plays a pivotal role in regulating HER2 overexpression in breast cancer cells [105]. Importantly, inhibition of PC-PLC hydrolysis activity using tricyclodecan-9-yl-potassium xanthate (D609) resulted in HER2 internalization and lysosomal degradation. Paris et al. showed that the combination of trastuzumab plus D609 resulted in significantly improved HER2 degradation relative to trastuzumab-treatment only and correlated it with improved cytotoxicity. Interestingly, D609 induced the SKOV3.ip cell line, which has 2-fold higher HER2 expression and 2.9-fold increased PC-PLC activity relative to the parental SKOV3 cell line, to have substantially increased HER2 internalization and lysosomal processing. Thus, PC-PLC is an attractive biomarker for combination anti-HER2 drugs plus D609. Further information on targeting phospholipid metabolism and clinical trials is reviewed in [106].

### 5.2. Receptor Crosslinking

Theoretically, using combinations of different antibodies or a single antibody specific for different binding sites can crosslink HER2 to form a large meshwork complex. In EGFR, this has been shown to induce the formation of a receptor/antibody cluster that then ‘collapses’ into the cell [107]. Importantly, the aspect of ‘collapsing’ into the cell was still receptor dependent and was associated with increased trafficking to lysosomes. Here, the evidence for HER2 is examined.

(A)Trastuzumab/T-DM1 plus pertuzumab cocktails

The current standard of care for the first-line treatment of HER2-positive metastatic breast cancer is a combination of trastuzumab plus a taxane and pertuzumab (Perjeta^®^). Pertuzumab is a mAb that targets domain II of HER2. Currently, pertuzumab is approved for use in combination with trastuzumab and docetaxel for patients with early and metastatic HER2-positive breast cancer [17]. There are several proposed mechanisms for combined trastuzumab-pertuzumab therapy and are reviewed in [108]. Franklin et al. generated a crystal structure of a Fab fragment of pertuzumab bound to the HER2 extracellular segment [109]. The orientation of pertuzumab binding to domain II of HER2 suggested that a reason for the increased cytotoxicity of pertuzumab is that dimer blocking extended to cells with relatively reduced HER2 expression levels, which improved the effectiveness of trastuzumab [109] (Figure 4). Pertuzmab also shows the ability to block HER2/EGFR dimerization to increase the contribution to trastuzmab. However, this effectiveness to kill cells ranges from minimal to effective in a panel of ovarian tumor cells with variable differences in EGFR and HER2 expression levels [110]. Although trastuzumab and pertuzumab undoubtedly cooperate with each other to improve tumor killing, mechanisms related to receptor crosslinking and internalization are scant. This section explores the literature for increased antibody internalization due to the combined binding of trastuzumab and pertuzumab or other antibody cocktails to HER2.

As previously mentioned, the ErbB family of receptor tyrosine kinases has four members: EGFR, HER2, HER3, and HER4. Several cancer types express elevated levels of EGFR relative to normal tissues, which is associated with reduced recurrence-free and overall survival rates [112]. Increased expression of HER3 is also associated with several cancer types and linked to poor survival [113]. Surprisingly, elevated HER4 expression was associated with improved survival in breast cancer [114]. In addition, expression of HER3, HER4 and their ligand heregulin-4 was associated with improved survival in bladder cancer patients [115].

Using a model of stably transfected porcine aortic endothelial (PAE) cells overexpressing HER2 only or HER2 and HER3, Hughes et al. showed a modest increase in the amount of internalized antibodies in cells treated with trastuzumab plus pertuzumab [83]. The majority of antibody remained at the cell surface. This internalization pattern was also observed in SKOV3 cells. Interestingly, the highest amount of HER2 internalization and degradation was in HER2/HER3-positive cells relative to HER2-only positive PAE cells. As described in Section 5.1, incubation with HSP90 inhibitors substantially increased the internalization and lysosomal degradation of trastuzumab. Hughes et al. demonstrated that the HSP90 inhibitor 17-AAG potentiated internalization of combined trastuzumab and pertuzumab. The most pronounced internalization and degradation occurred in HER2/HER3-positive, 7-AAG, and trastuzumab plus pertuzumab conditions [84].

Pertuzumab combined with T-DM1 has demonstrated synergistic cytotoxicity, increased anti-tumor potency, and clinical activity with an acceptable safety profile in a phase Ib/II study [116]. However, the MARIANNE phase III trial showed that T-DM1 plus pertuzumab for the treatment of patients with HER2-positive, progressive or recurrent locally advanced or metastatic breast cancer did not demonstrate superior PFS compared to trastuzumab plus a taxane [117,118]. Ultimately, the lack of evidence that pertuzumab plus trastuzumab or T-DM1, without support from HSP90 inhibition, increases internalization is a likely reason for why T-DM1 plus pertuzumab did not have increased anti-tumor activity in patients.

(B)Other anti-HER2 antibody cocktails

The concept of using multiple antibodies to simultaneously target HER2 has been in existence for nearly three decades and there is convincing evidence of increased internalization. Early studies from Kasprzyk et al. and Harwerth et al. first showed that two anti-HER2 mAbs increased HER2 degradation [66,80]. SKBR3, SKOV3, and N87 cells were pulse-labeled with ^35^S-cysteine and then chased with cold cysteine in the presence of each mAb alone or in combination. Western blot using an anti-HER2 antibody revealed that HER2 was absent in the exposed gel where it was present at the same level from cells treated with individual mAbs relative to untreated cells. This data was the first to suggest that two different mAbs specific to HER2 can bind, be internalized and degraded. Importantly, these studies suggested that a two-antibody cocktail binding distinct extracellular regions of HER2 have the ability to crosslink the receptor.

Hurwitz et al. determined that mAbs that bind HER2 to certain regions underwent internalization and had associated anti-tumor activity and other mAbs that bound to other regions did not internalize and had no anti-tumor activity [81]. Spiridon et al. generated a panel of mAbs against nine unique HER2 epitopes and demonstrated that a mixture of three mAbs binding unique epitopes induced hypercrosslinking of HER2 in BT474 cells [119]. It was Ben-Kasus et al. who first proposed the ‘lattice’ model for anti-HER2 crosslinking antibodies [120]. Non-epitope overlapping antibodies induce a large receptor aggregate, which is internalized and sorted for lysosomal degradation (Figure 5). Ben-Kasus et al. discovered that an effective combination of antibodies required that no antibody bind to a carbohydrate-containing epitope and that only one antibody in the mixture bind to the HER2 domain II, its dimerization domain. In addition, the mixture of the most potent combination of two antibodies showed that reduced antibody concentrations were internalized and caused HER2 degradation more effectively than at increased concentrations.

Pedersen et al. provided improved understanding and also introduced new complexities by identifying optimal epitope combinations that provide improved activity [49]. Novel mAbs for all four HER2 extracellular domains were tested in several HER2-positive tumor cell lines. Importantly, all the mAbs possessed high affinity ranging from 0.5 to 4.6 nmol/L and, as controls, pertuzumab and trastuzumab had affinities of 0.3 and 0.4 nmol/L, respectively. First testing dual mixtures, increased cytotoxicity was achieved when mAbs simultaneously targeted domains I + II, I + III, or III + IV in N87, HCC202, and BT474 cells. Interestingly, no dual mixture was effective against SKBR3 cells. In addition, there was no increased cytotoxicity in MDA-MB-175-VII and MCF7 cells that are driven by HER2/HER3 heterodimer-induced signaling. However, a three mAb cocktail had a profound cytotoxic improvement in several cell lines relative to dual mixtures. The most efficacious mixture included mAbs that targeted domains I, II, and IV. For the first time, Pedersen et al. discovered that domain I in combination with domains IV or II and IV was critical for internalization and subsequent degradation (Table 4). In resistance models where EGF and heregulin were added to the media containing N87 cells or OE19-derived clones, mAbs targeting domains I, II, and IV had superior cytotoxicity relative to trastuzumab plus pertuzumab. However, there were still variations in the level of cytotoxic potency in different cell lines treated with the three mAb cocktail. Overall, this study showed that targeting multiple specific domain combinations improved internalization and degradation and resulted in more effective anti-tumor activity relative to single mAbs or dual mixtures including the combination of trastuzumab plus pertuzumab. However, differences in responses most likely reflect differences in cancer cells or the level of HER2 expression. The relevance of HER2 expression was not explicitly addressed in the study. In addition, the level of HER2 crosslinking induced by two mAbs versus three mAbs was not evaluated.

(C)Bispecific antibodies/ADCs

Other strategies include the development of bispecific antibodies (BsAbs) that couple HER2 together with a receptor known to undergo rapid internalization. Andreev et al. hypothesized that because HER2 has limited internalization, if coupled to the prolactin receptor (PRLR) that is known to undergo rapid internalization [121], then anti-HER2 antibodies can have more efficient anti-tumor activities [75]. Testing the breast cancer cell line T47D, which has low levels of HER2 (4.00 × 10^4^ rec/cell) and even lower levels of PRLR (2.73 × 10^4^ rec/cell) expression, an anti-HER2 ADC (DM1 payload) was completely non-effective. In contrast, T47D cells were highly sensitive to the anti-PRLR ADC (DM1 payload). Indeed, a BsAb was able to bridge HER2 and PRLR and resulted in rapid internalization and lysosomal degradation in T47D cells. A HER2xPRLR BsAb ADC was dramatically more potent than the reference anti-HER2 and anti-PRLR ADCs.

Similarly, De Goeij et al. developed a HER2xCD63 BsAb ADC conjugated to the payload duostatin-3, a microtubule-disrupting agent [71]. CD63 is a member of the tetraspanin superfamily and is ubiquitously expressed [122]. The strategy was that CD63 is known to shuttle between the cell surface membrane and lysosomes and, hence, could facilitate ADC internalization, degradation, and duostatin-3 release. Testing HCC1954, SKOV3, and Colo205 cell lines that express high, intermediate, and low HER2 levels, respectively (Table 1). The BsAb HER2XCD63 had 4-fold greater internalization and lysosome processing rates compared to an anti-HER2 mAb. This result suggested that the HER2xCD63 BsAb-induced increased internalization was independent of HER2 expression levels. Although the increased cellular accumulation of the delivered duostatin-3 payload was not measured, cytotoxicity assays revealed that the HER2xCD63 BsAb ADC was more potent relative to the anti-HER2 parental ADC in SKOV3 and Colo205 cells, further suggesting that tumors with reduced HER2 expression can be effectively killed using this approach. Indeed, SKOV3 xenografts were effectively killed and tumor-bearing mice had increased survival to 105 days from the start of treatment. In contrast, anti-HER2 or anti-CD63 ADCs did not improve survival relative to mice treated with non-specific IgG (day 63).

(D)Biparatopic ADCs

Bispecific engineering technology has made an important advancement in the development of biparatopic (binding of different epitopes on the same antigen) antibodies and their development as anti-HER2 ADCs. Medimmune published results on a biparatopic ADC armed with a tubulysin derivative AZ13599185 [76]. The biparatopic antibody demonstrated the ability to induce extensive HER2 crosslinking that resulted in increased internalization, lysosomal localization, and HER2 degradation (Figure 5). However, examining a panel of cell lines with large variations in HER2 expression levels, the biparatopic antibody did not increase internalization in cell lines with low HER2 expression. Accordingly, cytotoxicity was limited in low expressing HER2-positive cells. Nonetheless, the biparatopic ADC was highly effective in tumor models representing different breast cancer patient subpopulations classified by HER2 expression including T-DM1 resistant models. The biparatopic ADC demonstrated an acceptable safety profile in cynomologous monkeys and recently completed a phase I/II trial (NCT02576548). Similarly, Zymeworks has developed an anti-HER2 biparatopic ADC and has an active phase I clinical trial (NCT03821233). There are other examples of biparatopic antibodies, such as the one consisting of pertuzumab genetically fused to a Fynomer scaffold targeting HER2 domain I showed the ability to increase internalization in N87 cells [123]. This result further supports the targeting of HER2 domain I as discovered by Pedersen et al. [49]. Interestingly, the Medimmune biparatopic antibody was constructed based on the trastuzumab and pertuzumab antibodies. Thus, the biparatopic antibody recognizes the same interactions sites and suggests that the limitations of trastuzumab plus pertuzumab can be overcome through the engineering of a single antibody capable of simultaneously binding these sites. It further suggests that the orientation differences of the biparatopic antibody also facilitate receptor crosslinking.

(E)Avidin/streptavidin-biotin system

In general, this system consists of a biotinylated anti-HER2 antibody that is allowed to bind to the surface of cells. In vitro, the cells would then be washed. In vivo, an appropriate amount of time for the antibody to clear and localize to the tumor would be required. Streptavidin is added to the cells or injected into the subject, which has the capacity to cluster HER2 by formation of extended crosslinks between HER2:Ab-biotin complexes. Moody et al. complexed biotin to fluorescently-tagged trastuzumab and evaluated internalization in SKBR3 and BT474 cells [124]. Only in the presence of streptavidin, were trastuzumab antibodies detected in lysosomes. In contrast, Zhu et al. showed this approach could not crosslink and internalize in HER2 low expressing MCF7 cells [125]. Interestingly, Wymant et al. demonstrated that the biotin/streptavidin-induced crosslinking of HER2 also induced accompanying degradation of HER3 but not EGFR. Whether additional receptors, if any, are complexed in the HER2 clusters and the effect on internalization remain a mystery. In addition, one should be cautious of the human immune system triggering a response to an avidin/streptavidin-biotin system.

### 5.3. Membrane Traversal Technologies

Cell-penetrating peptides (CPPs) have held great promise for developing next-generation pharmaceutics for their ability to effectively cross cell membranes and to deliver molecular cargoes and accumulate them in cells [126]. However, CPPs have been shown to abrogate antibody specificity [127]. There have been contradictory results when CPPs have been conjugated to anti-HER2 antibodies in combination with gold nanoparticles for their ability to specifically target and accumulate in HER2-positive cells. Cruz et al. attached trastuzumab to gold nanoparticles in combination with attached CPP HIV TAT [128]. Unfortunately, the CPP–gold conjugates had significant increased uptake in SKBR3 and MCF7 cells compared to CPP–gold–trastuzumab conjugates. In contrast, Guo et al. screened a panel of various anti-HER2 mAbs conjugated to the CPP penetratin [129]. One penetratin-anti-HER2 mAb retained HER2 specificity and efficiently internalized into MCF7 and PC-3 (HER2-positive) cells. When the penetratin-anti-HER2 mAb was complexed with gold, it resulted in increased cytotoxicity when combined with x-ray radiation. This work demonstrated that an agent with increased HER2-specific internalization could be utilized as a radiosensitizer.

Beaudoin et al. demonstrated that a nuclear localization signal (NLS) sequence containing peptide coupled with cholic acid (ChAcNLS) conjugated to T-DM1 enabled endosome escape and subsequent efficient nuclear localization in trastuzumab-resistant JIMT-1 cells [130]. Bile acids such as cholic acid are mainly known for their role of breaking down dietary lipids [131], but in the field of virology, they are known as essential host elements utilized by viruses to escape endosome entrapment [132,133]. Non-enveloped viruses that cannot rely on membrane fusion utilize bile acids to trigger endosome escape. Once internalized, the *Calciviridae* family utilizes cholic acid to activate sphingomyelinase to cleave sphingomyelin and form ceramide. Increased amounts of ceramide destabilize membranes by forming channels or lipid flip-flop sufficient for traversal into the cytoplasm [134,135]. ChAcNLS-modified T-DM1 was able to escape endosomes through the formation of increased levels of ceramide in JIMT-1 cells. Once in the cytoplasm, the NLS from SV40 large T-antigen was recognized by nuclear transport proteins and shuttled into the nucleus. This resulted in an increase accumulation of T-DM1 and enhanced cytotoxicity by 100-, 50-, and 10-fold relative to T-DM1 in SKBR3, OE19, and JIMT-1 cells, respectively.

### 5.4. Recommendations for Future Internalization and Accumulation Studies

It is clear that there is wide variation in methods to evaluate anti-HER2 antibody and ADC internalization and subcellular distribution. Perhaps consideration of incorporating the following details can improve the clarity of the results and their meaning for future investigational anti-HER2 antibodies/ADCs.

The conjugation of dyes directly to the antibodies/ADCs under investigation that only become fluorescent upon internalization into the acidic environment of endosomes and lysosomes is an effective approach to measure lysosomal processing efficiency. Riedl et al. and Nath et al. both published well-described methods and dyes for anti-HER2 ADCs and antibodies, respectively [67,77]. This approach removes necessary washing and cell-stripping steps used by studies in this review. If possible, one could consider the strategy by Lee et al., which incorporated fluorescence resonance energy transfer dyes into the linker connecting the antibody to the drug [136]. Using this approach, one can track both the antibody and payload inside cells.Incorporating LC–MS/MS to quantify absolute drug metabolite concentration (nmol/L) in tumor cells. One might also consider acid washing the surface of the cells prior to lysing, so that only intracellular concentrations can be determined. This way, it would be possible to accurately ascertain cellular accumulation levels, especially in cells with poor HER2 internalization and high levels of HER2 expression on the cell surface.Evaluate intracellular drug concentrations or antibody internalization in a minimum of three cells lines that have high, intermediate, and low HER2 expression levels and correlate with the impact on cytotoxicity.Compare cellular accumulation and cytotoxicity relationship with tumor accumulation and tumor killing to determine whether there is a correlation between in vitro and in vivo.Consider the time points used for evaluating internalization/accumulation and cytotoxicity, so that they can be directly associated based on time.Consider measuring the expression levels of EFGR, HER3, and HER4 and whether changes in internalization occur, including in the presence of activating ligands.

## 6. Future

Receptor-mediated antibody/ADC internalization and lysosomal processing are key mechanisms underlying several investigational anti-cancer agents. The HER2 system described in this review holds several insights for the design and evaluation of future antibodies and ADCs targeting HER2 and different cell surface receptors. These insights can also potentially aid in the development of aptamers targeting HER2, as these RNA-based agents, due to their high affinity, specificity, and non-immunogenicity, have been referred to as ‘chemical antibodies’ [137]. The most recently approved ADC sacituzumab-govitecan (SG; Trodelvy^®^) utilizes a linker containing several polyethylene glycol moieties that enables it to load twice the amount of drug per antibody than any currently approved ADC [138]. The drug payload SN-38 is a topoisomerase inhibitor. Although SG does not target HER2, this new ADC design to load more drugs onto the antibody in order to deliver more payload inside the cell underscores the importance of cellular accumulation and its importance for next-generation ADCs. Regarding the previously described T-DXT, reviews have highlighted the importance of internalization for this ADC, especially since its payload needs to localize to the nucleus [139,140]. However, none of the preclinical studies describe internalization or payload cellular accumulation [141,142,143]. After the challenges learned about HER2 internalization, the question is what aspect enabled the success of T-DXT? T-DXT targets the nucleus delivering the topoisomerase I inhibitor deruxtecan [144]. Is it because deruxtecan is more cytotoxic than DM1 and, hence, a reduced intracellular concentration is sufficient to evoke effective cytotoxicity? Hopefully future studies will address these points.

## 7. Conclusions

Although HER2 preferential overexpression on the surface of tumor cells is attractive, it is currently accepted that HER2 suffers from poor internalization. Furthermore, poor internalization and subsequent poor cellular accumulation, in the case of T-DM1, contribute to poor therapeutic effectiveness. These anti-HER2 agents are expensive. As a result, they have been questioned for their cost-effectiveness. This review examined the pertinent evidence and concludes that HER2-positive tumor cells that are invasive and that have relatively low expression are the most attractive for internalizing anti-HER2 antibody therapeutics. In addition, crosslinking using ADC cocktails or bispecific ADCs appears to be promising in increasing internalization and payload accumulation, which results in greater anti-tumor efficacy. Lastly, carefully planned in vitro studies to examine internalization/lysosomal processing and to quantify cellular accumulation will considerably aide the development of future investigational antibody therapeutics.

## 8. Patents

J.L. holds patent WO 2017/156630 A1 described in [130].

## Figures and Tables

**Figure 1 antibodies-09-00032-f001:**
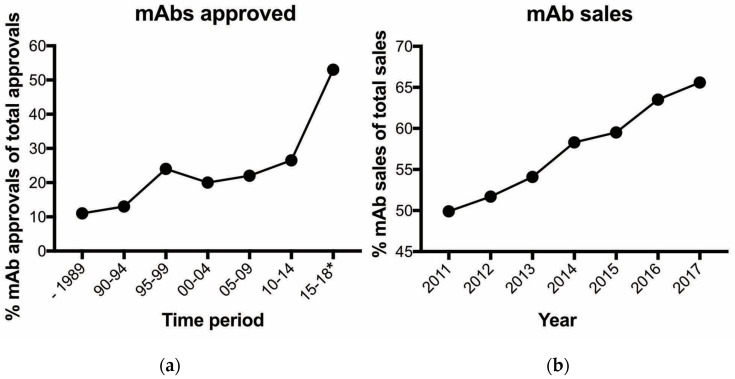
Overview of antibody impact in the biopharmaceutical market. (**a**) The percentage of first-time mAb approvals of total biopharmaceuticals during the indicated time periods. (**b**) The percentage of global mAb sales of total biopharmaceuticals for indicated years. Data obtained and adapted from reference [5]. * Until July 2018.

**Figure 2 antibodies-09-00032-f002:**
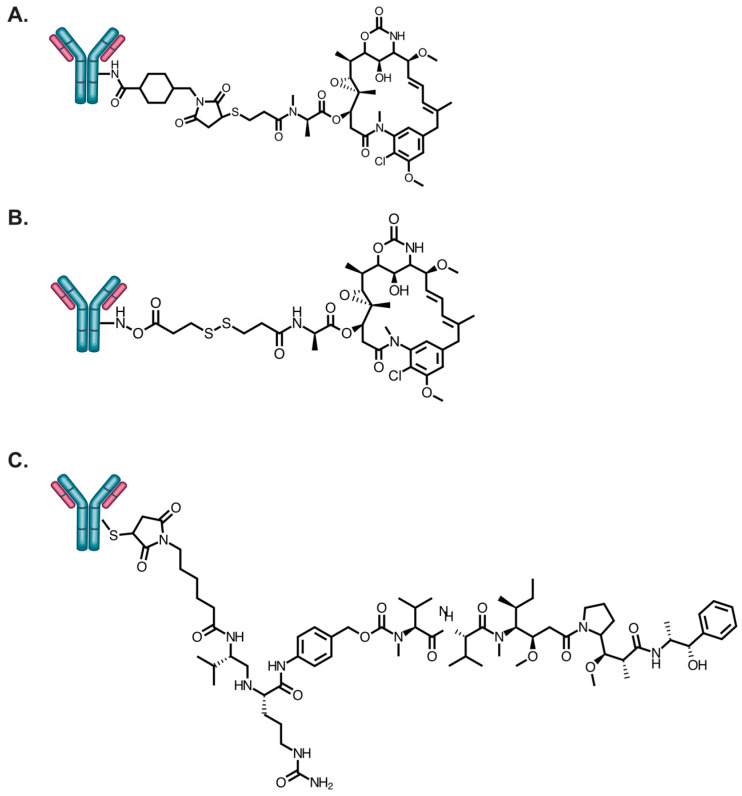
Structures of trastuzumab-based ADCs evaluated for cellular payload accumulation. (**A**) T-DM1, (**B**) T-SPP-DM1, and (**C**) T-vc-MMAE.

**Figure 3 antibodies-09-00032-f003:**
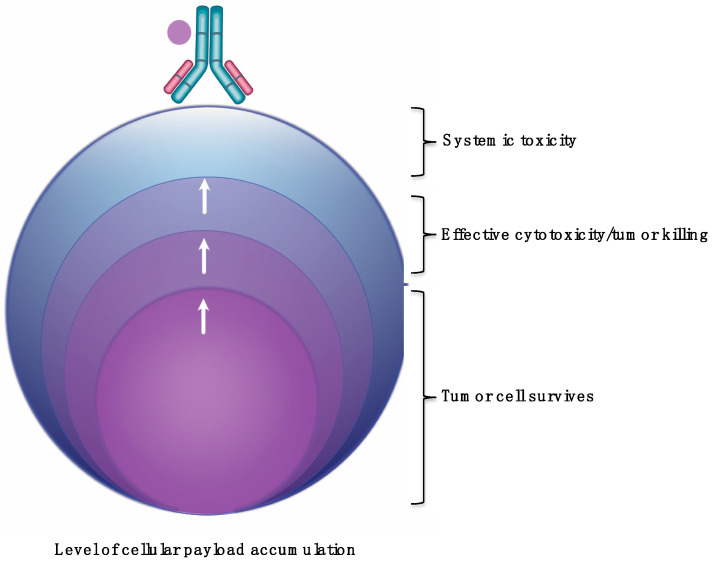
The working ‘goldilocks’ model of the required amount of cellular accumulation of anti-HER2 ADC-delivered payloads to evoke effective tumor killing. Too much drug accumulation causes systemic toxicity. Too few drugs accumulated results in tumor cell survival. An effective amount accumulates the drug at just the right concentration.

**Figure 4 antibodies-09-00032-f004:**
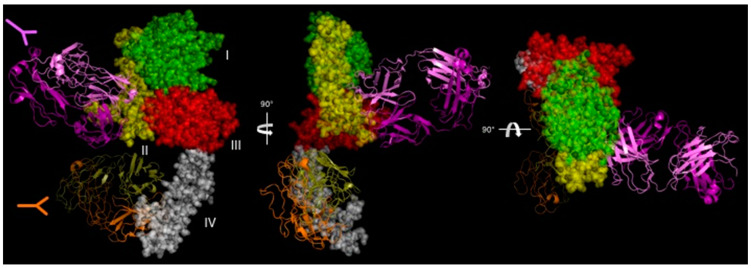
HER2 extracellular segment bound by the Fab of pertuzumab (purple/violet) and trastuzumab (orange/deep olive). HER2 domains I, II, III, and IV shown in green, yellow, red, and white, respectively. Full antibodies for pertuzumab and trastuzumab depicted by Y figures in violet and orange, respectively. Figure adapted from crystal structure 6OGE [111].

**Figure 5 antibodies-09-00032-f005:**
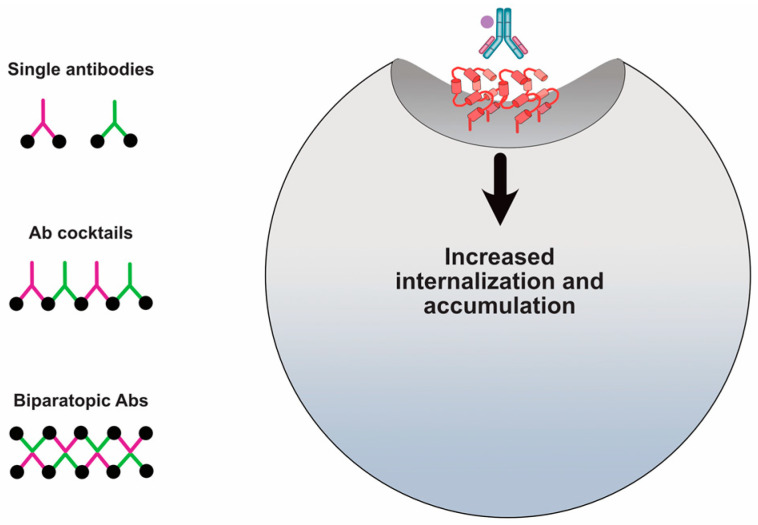
Schematic of different abilities to crosslink HER2 receptors, which results in increased internalization and antibody or payload cellular accumulation.

**Table 1 antibodies-09-00032-t001:** Cell lines based on HER2 expression levels studied for internalization and accumulation.

Cell Line	Source ^1^	Tumor Type ^2^	HER2 Expression ^3^	Internalization Studies	Accumulation Studies
Breast cancer					
SKBR3	PE	AC	High [52,63]	[44,46,47,49,51,52,58,64,65,66,67,68]	[69]
AU565	PE	AC	High [70]	[57]	
BT474	PT	IDC	High [52]	[44,49,52]	[69]
MCF7	PE	IDC	Low [70]	[44,54,67]	[69]
HCC1419	PT	Duc.Ca	High [52]	[52,68]	
HCC1954	PT	Duc.Ca	High [71]	[52,64,68]	
HCC2185	PE	MLCa	Low [52]	[52]	
HCC202	PT	Duc.Ca		[49]	
BT483	PT	IDC	Intermediate [52]	[52]	
CSC^4^	PE	IDC	Low [54]	[54]	
MDA-MB-175-VII	PE	IDC	Low [72]	[49]	
ZR75-1	PE	AC	Intermediate [52]	[52]	
KPL4	PE	AC	High/Int. [73]		[74]
T47D			Low [75]	[75,76]	
Prostate cancer					
LNCap	M	PC	Low [52]	[52]	
LAPC-4	M	PC	Low [52]	[52]	
C42B	M	PC	Low [52]	[52]	
Ovarian, gastric, colorectal cancers and melanoma					
SKOV3	AF	OC	Intermediate [48,71]	[47,48,66,77]	
SKOV3.ip	AF	OC	High [48,71]	[48]	
OE19	PT	OC	High [78]		[79]
N87	M	GC	High [80]	[49,80,81]	[82]
Colo205	PT	AC	Low [57]	[57]	
A431	PT	SC	Low [63]	[57,77]	
Non-tumor model systems					
Porcine aortic endothelial cells				[83,84]	

^1^ PE = pleural effusion; PT = primary tumor; M = metastasis; AF = ascitic fluid. ^2^ AC = adenocarcinoma; IDC = invasive ductal carcinoma; DucCa = ductal carcinoma; MLCa = metastatic lobular carcinoma; PC = prostate carcinoma; OC = ovarian carcinoma; GC = gastric carcinoma; SC = squamous carcinoma. ^3^ High ≥ 1 × 10^6^ rec/cell; intermediate >1 × 10^5^ to <1 × 10^6^ rec/cell; low ≤ 1 × 10^5^ rec/cell. References included. ^4^ A breast cancer CSC population was isolated with a low HER2 expression phenotype.

**Table 2 antibodies-09-00032-t002:** Study designs for quantitative determination of ADC payload cellular accumulation.

Study	LC–MS/MS Method	Cell and ADC	Cell Incubation Parameters
Erickson et al. [69]	Converted from radioactivity counts	BT474EEI, SKBR3, and MCF7 cells in T75 flasks (10^7^ cells). T-[^3^H]DM1 or T-SPP-[^3^H]DM1.	20–40 nmol/L ADC pulsed for 30 min. Media replaced with fresh media and sampled at 24 h.
Caculitan et al. [74]	Calibration curve	KPL4 and KPL4-cathepsin B knocked down cells in 96-well plates. T-MMAE with protease-sensitive vc linker.	0–10 μg/mL ADC for 24 h.
Caculitan et al. [74]	Calibration curve	KPL4 and KPL4-cathepsin B knocked down cells in T75 flasks. T-MMAE with vc linker or vc isomer that is protease-resistant (R).	1 h pulse. Media replaced with fresh media and sampled at 4 h or 24 h. ADC concentration not reported.
Sauveur et al. [79]	Calibration curve	OE19 and OE19-resistant cells exposed to T-DM1. Cell format not reported.	5 nmol/L for 24 h.
Wang et al. [82]	Calibration curve	N87 and N87-KR cells exposed to T-DM1 in 6-well plates.	10 μg/mL for 24 h.
Uppal et al. [92]	Calibration curve	750,000 donor cells separated into hematopoietic stem cells, immature and mature megakaryocyte populations.	6.25 μg/mL of T-DM1 for 24 h.

**Table 3 antibodies-09-00032-t003:** Relationship between cell type, and payload accumulation and cytotoxicity.

Non-Cleavable Trastuzumab ADCs				
Cells	ADC	T-DM1 Sensitive	Accumulation (nmol/L)	IC_50_ (nmol/L)
KPL4	T-vc(R)-MMAE	Yes	125	0.42
SKBR3	T-DM1	Yes	500	0.0073
BT474EEI	T-DM1	Yes	300	0.04
MCF7	T-DM1	Yes	500	0.02
OE19	T-DM1	Yes	4.1 ^1^	0.05
OE19TR	T-DM1	No	5.7 ^1^	0.73
OE19TCR	T-DM1	No	2.5 ^1^	0.98
N87	T-DM1	Yes	4.9 ^1^	0.2
N87KR	T-DM1	No	3.2 ^1^	12.5
Premature megakaryocytes ^2^	T-DM1	Yes	8.2	20
Cleavable/reducible trastuzumab ADCs				
KPL4	T-vc-MMAE	Yes	475	0.2
BT474EEI	T-SPP-DM1	Yes	200	0.2
MCF7	T-SPP-DM1	Yes	150	0.07

^1^ Values provided were converted into nmol/L. ^2^ This study found that intracellular access and accumulation were not HER2 mediated.

**Table 4 antibodies-09-00032-t004:** Antibody cocktail effect on internalization and HER2 degradation from [49] ^1^.

Domains	Internalization	Relative HER2 Degradation ^2^
I and II	No	No
II and IV	No	Weak
I and IV	Yes	Strong
I, II, and IV	Yes	Strong
Trastuzumab + Pertuzumab	Yes	Weak

^1^ Single mAbs, including trastuzumab and pertruzumab, did not show internalization and HER2 degradation in SKBR3, BT474, OE19, and HCC202 cells. ^2^ Relative to untreated and control Ab-treated cells.

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
