# Peer review of "Improving Receptor-Mediated Intracellular Access and Accumulation of Antibody Therapeutics—The Tale of HER2"

_2073-4468, 2020, doi:10.3390/antib9030032_

Round 1

Reviewer 1 Report

Research on internalization of HER2 and/or anti-HER2 antibodies is a field with contradicting reports and opinions.  In this comprehensive review Leyton covers these questions  in a balanced way and gives an updated view on how, among others, the use of different cell lines might be a likely reason for some of the controversies that exists.  The review is well written and organized, with only a very few, minor spelling mistakes.  If anything should be criticized, the title might be a bit misleading  since the paper deals with HER2 and anti-HER2 trafficking not only as a model, but as the main subject. To what extent the reviewed data can be transferred to the improvement of receptor-mediated intracellular access and accumulation of antibody therapeutics in general , is not directly clear. A bit beside the topic, but clearly related, is the use of aptamers  and aptamer-conjugated drugs. It is not an absolute request, but  a short description of this would make the review even better.

Author Response

Reviewer #1

Research on internalization of HER2 and/or anti-HER2 antibodies is a field with contradicting reports and opinions.  In this comprehensive review Leyton covers these questions in a balanced way and gives an updated view on how, among others, the use of different cell lines might be a likely reason for some of the controversies that exists.  The review is well written and organized, with only a very few, minor spelling mistakes.  If anything should be criticized, the title might be a bit misleading since the paper deals with HER2 and anti-HER2 trafficking not only as a model, but as the main subject. To what extent the reviewed data can be transferred to the improvement of receptor-mediated intracellular access and accumulation of antibody therapeutics in general, is not directly clear. A bit beside the topic, but clearly related, is the use of aptamers and aptamer-conjugated drugs. It is not an absolute request, but a short description of this would make the review even better.

Response. Thank you for appreciating the viewpoint from which I wrote this review. HER2 is an important target for antibody therapeutics in oncology. I aimed for an objective and comprehensive review that for the first time provides in a single document clarity on HER2 internalization and its impact on antibody-based therapeutics specifically related to intracellular access and accumulation.

I agree that the title is a bit misleading. Since HER2 is arguably the most clinically impactful target for antibodies, my original intent was that this review could propel researchers studying different tumor antigens based on what they read and learned in this review using HER2 as a framework model.  However, I agree that this is too presumptive and have changed the title to “Improving receptor-mediated intracellular access and accumulation of antibody therapeutics – The tale of HER2” as I believe this states the sole focus will be on HER2 but subtly hints at impact on other receptor systems.

I agree that aptamers important. In addition, I found numerous reports of aptamers targeting HER2 for diagnostic and therapeutic applications. One review states “Compared with antibodies, aptamers have such advantages as high specificity and affinity, stability, and non-immunogenicity. In addition, they can be easily modified at a low cost to target a variety of molecules and are, therefore, called “chemical antibodies””. I am not an aptamer expert and am hesitant to describe in detail the mechanistic underpinnings of their therapeutic action. Furthermore, it would be an enormous task to investigate aptamers and what happens when they get internalized. Its unclear if their mechanistic action is empowered by internalization to which, this review is about. Lastly, because aptamers are not antibodies I feel this would take away from the focus of the review. Instead, I add a sentence mentioning the potential power of aptamers in the section 6. Future and place a reference. I hope you agree that this can lead readers to the wonderful world of aptamers without moving too far away from the focus on antibodies.

Reviewer 2 Report

In this manuscript, Dr Leyton reports an interesting overview of the current knowledge and strategies to enhance antibody therapeutics HER2 mediated internalization and accumulation. After the introduction, a much appreciated section (part 2) introduces the question of antibody therapeutics cost-effectiveness, justifying the importance of the review’s topic. In a third part, the author describes the current knowledge on HER2 mediated antibody therapeutics internalization, eventually pointing out the still ambiguous relationship between internalization and cytotoxicity of antibody-based therapeutics. This logically leads to the fourth part in which the author discusses the relationship between ADC drug cellular accumulation and cytotoxicity. In the fifth part, he presents the approaches which have been proposed to improve both anti-HER2 antibodies internalization and accumulation. Finally, before the general conclusion, he makes a few brief remarks about two recently approved ADCs.

General Remarks:

Globally, the topic of the manuscript is of high interest to the field of antibody-based therapeutics. The manuscript is soundly organized and overall well-written, although some adjustments should be made to improve the clarity of certain passages.

The manuscript presents studies on internalization and accumulation of HER2-targeting agents, but the title and abstract state that the HER2 framework will be used as a model. This implies the findings will not only be discussed but ultimately used to give perspective on agents targeting other antigens than HER2. A last section in which the author compares HER2 to other antigens, and shows the significance of the presented findings beyond HER2 targeting, would significantly improve the interest of the review.

Improvements:

57: « Carter and Lazar » needs a citation

68-69: The lines 794-802 would be more relevant if placed here rather than in the conclusion

226-258: The authors list evidence that internalization is favored by low levels of HER2 expression. The sections 243-247 and 248-253 seem to be in contradiction regarding the impact of heterodimer formation on antibody internalization, which should be further discussed. In 254-256, the authors should develop this part to explain how the impact of membrane smoothness on internalization is linked to the HER2 expression impact on internalization.

274-275: The fact that Phillips’ and Erickson’ findings regarding T-DM1 toxicity towards MCF7 cells are diametrically opposed should be further commented upon.

283-284: In the text, the terms “lipophilic” and “hydrophilic” must be switched since in the cited article, DDAO is non-residualizing and lipophilic, while IRDye800CW is hydrophilic and residualizing.

356: In Table 2, references numbers should be included (as they are in Table 1).

357-383: The clarity of this paragraph could be improved from being restructured so that the in vitro and in vivo results are clearly separated.

376: Rephrase to “Plasma clearance of T-DM1 was ~2-fold slower than that of T-SPP-DM1”, so that the adjective “slower” is attributed the plasma clearance and not to the compounds themselves.

393: Linkers that result in a ‘clean’ release of the drug are commonly referred to as “traceless linkers”. This term should be mentioned so that readers who are interested can search for it.

414-416: Here, the term “weaker” (used twice) should be replaced by “higher” when used to qualify values of IC50, to avoid confusions.

441-458: The fact that one cell line lacks HER2 expression calls into question the relevance of this study in the context of this review.

519: Figure 3 might not be necessary.

539: “PC-PLC inhibition of its hydrolysis activity” should be rephrased to “inhibition of PC-PLC hydrolysis activity”

599-601: A “lack of clear evidence” cannot be “a potential reason” for a treatment inefficiency.

Minor corrections:

97: The strategy to to conjugate

164: replace “trafficking to T-DM1” by “trafficking of T-DM1”

165: replace “each on its own” by “each on their own”

165: replace “result in T-DM1 unable to” by “result in T-DM1 being unable to”

174: replace “as the evidence reveals” by “as the evidence reveal” (if “evidence” is meant to be used as a plural)

288: replace “analyzed” by “analysis”

340: replace “expression levels can be found” by “expression levels, which can be found”, or rephrase.

361: replace “T-DPP-DM1” by “T-SPP-DM1”

379: replace “This study indicated indicated” by “This study indicated”

384: In figure 2. C., the structure of MMAE could benefit from a “Clean up structure” to correct the bonds’ angles.

400-401: replace “accumulated ~450 nmol/L ~125nmol/L MMAE catabolites” by “accumulated ~450 nmol/L and ~125 nmol/L of MMAE catabolites”

444: replace “12.9&” by “12.9%”

455: replace “control ADCs were equally cytotoxic potency” by “control ADCs had equal cytotoxic potency”

475: replace “demonstrated that that” by “demonstrated that”

484: replace “MCF7 as the only” by “MCF7 was the only”

545: replace “intrnalization” by “internalization”

559/560: replace “and are reviewed in” by “and they are reviewed in” or “which are reviewed in”, or rephrase.

563: replace “relative reduced” by “relatively reduced”

578: replace “and is associated” by “which is associated”, or rephrase.

610: replace “can bind be internalized” by “can bind, be internalized”

645: replace “HER2 expression as not” by “HER2 expression was not”

670: replace “(Table 1), The BsAb” by “(Table1), the BsAb”

733: replace “subsequent efficiently nuclear localization” by “subsequent efficient nuclear localization”

738: replace “sphnigomyelinase” by “sphingom

Author Response

Reviewer #2

In this manuscript, Dr Leyton reports an interesting overview of the current knowledge and strategies to enhance antibody therapeutics HER2 mediated internalization and accumulation. After the introduction, a much appreciated section (part 2) introduces the question of antibody therapeutics cost-effectiveness, justifying the importance of the review’s topic. In a third part, the author describes the current knowledge on HER2 mediated antibody therapeutics internalization, eventually pointing out the still ambiguous relationship between internalization and cytotoxicity of antibody-based therapeutics. This logically leads to the fourth part in which the author discusses the relationship between ADC drug cellular accumulation and cytotoxicity. In the fifth part, he presents the approaches which have been proposed to improve both anti-HER2 antibodies internalization and accumulation. Finally, before the general conclusion, he makes a few brief remarks about two recently approved ADCs.

Response. Thank you for comments on the review structure. I sincerely also appreciate your comments on the section I decided to write on antibody cost-effectiveness. I think its important as scientists this is addressed and I don’t see much literature particularly for antibodies/ADCs on the subject. Based on some excellent points from Reviewer 3 this section is further expanded and hopefully very good to disseminate to the scientific community.

General Remarks:

Globally, the topic of the manuscript is of high interest to the field of antibody-based therapeutics. The manuscript is soundly organized and overall well-written, although some adjustments should be made to improve the clarity of certain passages.

The manuscript presents studies on internalization and accumulation of HER2-targeting agents, but the title and abstract state that the HER2 framework will be used as a model. This implies the findings will not only be discussed but ultimately used to give perspective on agents targeting other antigens than HER2. A last section in which the author compares HER2 to other antigens, and shows the significance of the presented findings beyond HER2 targeting, would significantly improve the interest of the review.

Response. Thank you for appreciating on the impact of this review to antibody scientists. I agree the title is misleading. Reviewer #1 shared your view. I repeat the answer I previously used to address your comment. Since HER2 is arguably the most clinically impactful target for antibodies, my original intent was that this review could propel researchers studying different tumor antigens based on what they read and learned in this review using HER2 as a framework model.  However, I agree that this is too presumptive and have changed the title to “Improving receptor-mediated intracellular access and accumulation of antibody therapeutics – The tale of HER2” as I believe this states the sole focus will be on HER2 but subtly hints at impact on other receptor systems.

Improvements:

57: « Carter and Lazar » needs a citation

Thank you for catching this and the reference is now inserted.

68-69: The lines 794-802 would be more relevant if placed here rather than in the conclusion

Thank you for the suggestion. You are right and that it is better to provide a brief concluding paragraph at the end of the Introduction so that the reader clearly sees what to anticipate from the review.

226-258: The authors list evidence that internalization is favored by low levels of HER2 expression. The sections 243-247 and 248-253 seem to be in contradiction regarding the impact of heterodimer formation on antibody internalization, which should be further discussed. In 254-256, the authors should develop this part to explain how the impact of membrane smoothness on internalization is linked to the HER2 expression impact on internalization.

Thank you for catching this and I agree there was a contradiction in the Valabrega et al., point and its place among the other studies is incorrect. I moved the Valabrega description to section 3.1 because it actually supports HER2 as not a very good internalizing receptor. The Valabrega paper developed its study based on the previously known observation that HER2 when it forms dimers with other HER family members, it stops their rapid internalization. Valabrega et al., discovered the link between HER2:ErbB heterodimers and resistance to trastuzumab. Ligand binding induced EGFR to have an ‘active’ conformation that enabled to form dimers preferentially with HER2 because HER2 was abundantly present due to its overexpression. The HER2:EGFR heterodimers reduced the available HER2 interaction sites for trastuzumab.

I reviewed de Goeij et al., and made the study findings clearer. Although the study also evaluates heterodimerization and the impact on trastuzumab internalization, the findings are in contrast to Valabrega et al. This study showed that antibodies (not trastuzumab) that do not interfere with HER2 heterodimerization internalize better that target HER2 monomers. Interestingly, heterodimer formation was enhanced when HER2 was expressed at low levels. For these reasons I keep de Goeij et al., for supporting the low HER2 expression enhances internalization finding, while moving Valabrega et al., to the section that HER2 is not a very good internalizing receptor.

I did further investigations on the phenomenon of membrane ruffling and added additional information in the point discussing the findings by Fehling-Kaschek et al. I include the link with HER2 expression and the finding was surprising. Unlike the previous finding, membrane areas with ruffling had high HER2 densities. In comparison, zones with smooth membranes had low HER2 densities. Thus, this study had increased complexity to HER2 internalization. I hope it is satisfactory for you.

274-275: The fact that Phillips’ and Erickson’ findings regarding T-DM1 toxicity towards MCF7 cells are diametrically opposed should be further commented upon.

Thank you for noticing this detail concerning MCF7 sensitivity to T-DM1. MCF7 is a very complicated cell line to describe. First, the literature is not clear on how to classify MCF7 with as either HER2-negative or HER2 low. It appears as a matter of convenience sometimes, when an anti-HER2 drug does not kill MCF7, the cell line is referred to as HER2 negative. Other times, MCF7 is described as HER2 low and this feature is used to emphasize that a novel anti-HER2 drug is very potent because it can potentially be used to treat patients who are 1+ or 2+ HER2 based on immunohistochemistry. I now insert in the opening paragraph of subsection 3.3 “For cells with HER2 high expression, poor internalization may be compensated by the excess amounts of HER2 at the cell surface that enough receptor still gets internalized. For cells with HER2 low expression, the efficient internalization may compensate for reduced amounts of HER2 at the cell surface”. In addition, I still include the differences between internalization and cytotoxicity assays. Typically, cytotoxicity assays are performed at much longer time points than internalization assays, hence, this allows more time for even poor internalizing cells to accumulate sufficient amounts of ADCs to evoke cytotoxicity.

283-284: In the text, the terms “lipophilic” and “hydrophilic” must be switched since in the cited article, DDAO is non-residualizing and lipophilic, while IRDye800CW is hydrophilic and residualizing.

Thank you for noticing this detail and it is now fixed.

356: In Table 2, references numbers should be included (as they are in Table 1).

The references are now included.

357-383: The clarity of this paragraph could be improved from being restructured so that the in vitro and in vivo results are clearly separated.

Thank you the comment. Yes, this was a bit of difficult. I originally wanted to dedicate a separate section on quantified drug accumulation in tumors. But there is very little literature that have done this for HER2. So I decided to present tumor uptake when it did and Erickson et al., was the only study that measured accumulation in vitro in cells and in vivo in tumors. I separated the Erickson et al., study into two paragraphs and hope it is clearer now. An outlier was the Zhang et al., study that quantified the amount of an anti-HER2 ADC with the PBD payload in tumors but did not quantify cellular accumulation. For this reason, I treat this study by itself and place it last in the section.

376: Rephrase to “Plasma clearance of T-DM1 was ~2-fold slower than that of T-SPP-DM1”, so that the adjective “slower” is attributed the plasma clearance and not to the compounds themselves.

The sentence now reads “Pharmacokinetic studies revealed that T-DM1 had a ~2-fold slower plasma clearance than T-SPP-DM1”.

393: Linkers that result in a ‘clean’ release of the drug are commonly referred to as “traceless linkers”. This term should be mentioned so that readers who are interested can search for it.

Thank you for the insight and ‘traceless’ is now included when I introduce the vc-linker.

414-416: Here, the term “weaker” (used twice) should be replaced by “higher” when used to qualify values of IC50, to avoid confusions.

I have replaced “higher” for “weaker”. Thank you for the correction.

441-458: The fact that one cell line lacks HER2 expression calls into question the relevance of this study in the context of this review.

Thank you for pointing this out. I absolutely agree. However, I found Uppal et al., study really interesting that cellular accumulation was quantified for healthy cells. Although the study started out assuming that the mechanism of internalization was HER2-mediated, the discovered it was not the case. I would like to keep this finding in the review because it it relevant for scientists working on anti-HER2 antibodies to the study links with safety and cellular accumulation. It is a one of a kind study.

519: Figure 3 might not be necessary.

I would very much appreciate it if I can keep Figure 3.

539: “PC-PLC inhibition of its hydrolysis activity” should be rephrased to “inhibition of PC-PLC hydrolysis activity”

This is now fixed.

599-601: A “lack of clear evidence” cannot be “a potential reason” for a treatment inefficiency.

I appreciate the correction in syntax. To keep things simple I removed ‘clear’ and replaced ‘potential’ for ‘likely’. 

Minor corrections:

97: The strategy to to conjugate

This is now fixed.

164: replace “trafficking to T-DM1” by “trafficking of T-DM1”

This is now fixed.

165: replace “each on its own” by “each on their own”

This is now fixed.

165: replace “result in T-DM1 unable to” by “result in T-DM1 being unable to”

This is now fixed.

174: replace “as the evidence reveals” by “as the evidence reveal” (if “evidence” is meant to be used as a plural)

This is now fixed.

288: replace “analyzed” by “analysis”

This is now fixed.

340: replace “expression levels can be found” by “expression levels, which can be found”, or rephrase.

This is now fixed.

361: replace “T-DPP-DM1” by “T-SPP-DM1”

This is now fixed.

379: replace “This study indicated indicated” by “This study indicated”

This is now fixed.

384: In figure 2. C., the structure of MMAE could benefit from a “Clean up structure” to correct the bonds’ angles.

I use ChemDoodle to draw chemical structures. I used the “optimize structures in 2D” function. The new chemical structure for vc-MMAE is now in Fig.2C. I hope it is satisfactory.

400-401: replace “accumulated ~450 nmol/L ~125nmol/L MMAE catabolites” by “accumulated ~450 nmol/L and ~125 nmol/L of MMAE catabolites”

This is now fixed.

444: replace “12.9&” by “12.9%”

This is now fixed.

455: replace “control ADCs were equally cytotoxic potency” by “control ADCs had equal cytotoxic potency”

This is now fixed.

475: replace “demonstrated that that” by “demonstrated that”

This is now fixed.

484: replace “MCF7 as the only” by “MCF7 was the only”

It is supposed to be “as” and not “was”.

545: replace “intrnalization” by “internalization”

This is now fixed.

559/560: replace “and are reviewed in” by “and they are reviewed in” or “which are reviewed in”, or rephrase.

This is now fixed.

563: replace “relative reduced” by “relatively reduced”

This is now fixed.

578: replace “and is associated” by “which is associated”, or rephrase.

This is now fixed.

610: replace “can bind be internalized” by “can bind, be internalized”

This is now fixed.

645: replace “HER2 expression as not” by “HER2 expression was not”

This is now fixed.

670: replace “(Table 1), The BsAb” by “(Table1), the BsAb”

We replaced the comma with a “.”

733: replace “subsequent efficiently nuclear localization” by “subsequent efficient nuclear localization”

This is now fixed.

738: replace “sphnigomyelinase” by “sphingom

This is now fixed.

Reviewer 3 Report

This manuscript provides a review of antibody internalization mechanisms in the context of HER2 expression in tumor cells, and the impact on antibody-drug therapeutics based on HER2 targeting. A comprehensive description of internalization pathways and impact of HER2 binding, complexing and efficacy of HER2 antibodies and ADCs is provided. 

The overview of economics of HER2 antibodies (Section 2) is interesting, but does not adequately address the approvals of T-DM1 in many western countries where health economics are part of the approval process. Nor is trastuzumab-deruxtecan included, which in view of recent approval in the US should be addressed. To justify this section within the review, more detail and context to other biologics would be required.

In Section 3, an introduction to HER2 at the start of the section, rather than towards the end of the section, is suggested to improve reader understanding of the structure and biology of HER2.

Section 4 summarises a number of papers describing the internalization of HER2-ADCs, and impact on cell killing. This section is too descriptive, and summaries of the key findings of relevant papers, with perhaps a Figure to assist with an overview of this section is strongly suggested. Moreover, data on HER2-ADCs other than T-DM1 should be provided.

In section 5, there is not compelling data presented to support Pertuzumab increasing T-DM-1 internalization, and therefore does not support this mechanism of action for the enhanced cell killing of the combination in models and clinical trials. Also, Avidin and Streptavidin-biotin systems have been shown to be immunogenic in humans, and not suited for clinical use - this should be indicated in the text. All strategies to enhance internalization should have clear reference to the feasibility of human use for context to the proposed strategies, and a Table would assist with this justification.

Author Response

Reviewer #3

This manuscript provides a review of antibody internalization mechanisms in the context of HER2 expression in tumor cells, and the impact on antibody-drug therapeutics based on HER2 targeting. A comprehensive description of internalization pathways and impact of HER2 binding, complexing and efficacy of HER2 antibodies and ADCs is provided. 

The overview of economics of HER2 antibodies (Section 2) is interesting, but does not adequately address the approvals of T-DM1 in many western countries where health economics are part of the approval process.

Thank you for pointing this out and I have re-investigated this area. You are right, in Western countries the process of “whether to reimburse or not” for countries with universal healthcare is highly complex and is evaluated on a case per case basis. So I cannot describe a general process for T-DM1. Hence, it was my decision to focus on the process undertaken in England as this was the most interesting. In brief, the National Institute for Heath and Care Excellence (NICE) rejected T-DM1 on the grounds that the drug was not cost-effective. I detailed how Roche and NICE went through the health economic evaluation process. To reflect on how health economic evaluations is undertaken in the US (the only Western country with private healthcare) I examined Le et al. Ultimately, both studies support each other. Similar studies in Canada, Australia, and Spain also support NICEs evaluation of T-DM1 cost-effectiveness. I believe my rewrite of the section is now very strong.

Nor is trastuzumab-deruxtecan included, which in view of recent approval in the US should be addressed. To justify this section within the review, more detail and context to other biologics would be required.

Thank you for suggesting this. Yes, we are in an exciting time with the approval of trastuzumab-deruxtecan just this past December. I have included a description of trastuzumab-deruxtecan due to its timeliness as being a just approved anti-HER2 ADC. I briefly describe its success in clinical trial leading to its accelerated approval in the US. Since the drug is so new there is no health economic evaluations on it. However, I did find that the drug was priced very high and is expensive, similar to T-DM1. Thus, I found and mentioned that England has now scheduled a health economic review of trastuzumab-deruxtecan. Please also note that I found no studies that investigated internalization or accumulation of trastuzumab-deruxtecan. This is why you do not see it in the rest of my review. However, I do elude to this fact in Section 6 emphasizing that these studies should be performed after the lessons learned with the difficulty of HER2 internalization. 

Regarding other biologics, the focus of my review is on antibody/antibody-based agents. The field is very large and would be beyond the scope of this review and the journal Antibodies if I were to include every anti-HER2 biologic.

In Section 3, an introduction to HER2 at the start of the section, rather than towards the end of the section, is suggested to improve reader understanding of the structure and biology of HER2.

Paragraph describing HER2 is now moved to the front of Section 3.

Section 4 summarises a number of papers describing the internalization of HER2-ADCs, and impact on cell killing. This section is too descriptive, and summaries of the key findings of relevant papers, with perhaps a Figure to assist with an overview of this section is strongly suggested. Moreover, data on HER2-ADCs other than T-DM1 should be provided.

Thank you for your comments. I intentionally wrote Section 4 very descriptively. Quantification of cellular payload accumulation and the impact on tumor killing is a major underlying aspect that all ADC scientists agree on, yet there is no comprehensive analysis of this area. My description brings the field up to speed in not only findings but on the methods used. Section 4 also includes Tables 2 and 3 that describe the differences in the LC-MS/MS methods used and the Accumulation results and impact on cytotoxicity, respectively.  Section 4 also includes Figure 3 that is a nice schematic that summarizes the accumulation findings.

With regards to HER2-ADCs other than T-DM1 should be provided, as described in the opening paragraph I searched the literature for “any” ADC that quantified payload accumulation in the HER2 system and all but one study were with trastuzumab-based ADCs or T-DM1. This underlies the importance and signals to future investigational ADCs (non-trastuzumab-based) to study cellular accumulation. To clarify this for readers I now write it clearly that the studies found that quantified cellular accumulation used T-DM1 or trastuzumab-based ADCs.

In section 5, there is not compelling data presented to support Pertuzumab increasing T-DM-1 internalization, and therefore does not support this mechanism of action for the enhanced cell killing of the combination in models and clinical trials.

Thank you for your comments. I believe the trastuzumab/pertuzumab cocktail is very relevant for this section. It was the “original” cocktail of two clinically validated independent antibodies. One reason people thought trastuzmab+pertuzumab worked was due to increased crosslinking and internalization but evidence for this was scant. My review for the first time puts together the evidence and shows that trastuzumab+pertuzumab does not increase internalization and provides an explanation for why T-DM1+pertuzumab failed in clinical trial. With all due respect, I have decided to keep the story of trastuzumab+pertuzumab as part A in Section 5.2, because it is vital as a lead-in for part B, which demonstrates that other cocktails that target other HER2 domains can successfully increase internalization.

Also, Avidin and Streptavidin-biotin systems have been shown to be immunogenic in humans, and not suited for clinical use - this should be indicated in the text.

I have indicated in the text that scientists should be cautious moving forward with avidin-biotin systems as they can cause human immune systems to trigger a response.

All strategies to enhance internalization should have clear reference to the feasibility of human use for context to the proposed strategies, and a Table would assist with this justification.

Thank you for the comment. In my review I do include personal perspectives namely, 1) drug affordability for patients and healthcare systems is important for antibodies/ADCs that are not cost-effective; 2) A scientific approach to make antibodies/ADCs more cost-effective is to make them more clinically effective by increasing their intracellular access/payload accumulation; and 3) there is no consensus on HER2 internalization, accumulation studies or methods, which for the first time my review brings to light for the entire research community. Humbly, I have included sufficient ‘viewpoints’ and do not feel the need to include my perspectives on whether or not the technologies presented in Section 5 will succeed in the clinic or not. I think the individual parts in Section 5 are concisely written so any reader can quickly access the references if they wish to learn more.
